# Inference-Time Reward Hacking
# in Large Language Models

**Hadi Khalaf**
Harvard University

**Claudio Mayrink Verdun**
Harvard University

**Alex Oesterling**
Harvard University

**Himabindu Lakkaraju**
Harvard University

**Flavio du Pin Calmon**
Harvard University

## Abstract

A common paradigm to improve the performance of large language models is optimizing for a reward model. Reward models assign a numerical score to an LLM's output that indicates, for example, how likely it is to align with user preferences or safety goals. However, reward models are never perfect. They inevitably function as proxies for complex desiderata such as correctness, helpfulness, and safety. By overoptimizing for a misspecified reward, we can subvert intended alignment goals and reduce overall performance – a phenomenon commonly referred to as reward hacking. In this work, we characterize reward hacking in inference-time alignment and demonstrate when and how we can mitigate it by *hedging* on the proxy reward. We study this phenomenon under Best-of-$n$ (BoN) and Soft Best-of-$n$ (SBoN), and we introduce Best-of-Poisson (BoP) that provides an efficient, near-exact approximation of the optimal reward-KL divergence policy at inference time. We show that the characteristic pattern of hacking as observed in practice (where the true reward first increases before declining) is an inevitable property of a broad class of inference-time mechanisms, including BoN and BoP. To counter this effect, we introduce `HedgeTune`, an efficient algorithm to find the optimal inference-time parameter. We demonstrate that hedging mitigates reward hacking and achieves superior reward-distortion tradeoffs on math, reasoning, and human-preference setups.

## 1 Introduction

Almost all current alignment methods, including BoN [1, 2], RLHF [3, 4], DPO [5], and their variants, aim to maximize a reward function while minimizing divergence from the original model's outputs [1]. It is important to distinguish between two types of rewards: proxy rewards which are the computable signals we directly use during alignment (like scores from a trained reward model), and true rewards, which represent the often latent quality of the model's output according to a desired objective. As the name suggests, proxy rewards are approximations of the true reward and, consequently, of intended alignment goals such as correctness, helpfulness, and safety.

A fundamental challenge persists across reward-based alignment methods: **all proxy reward models are imperfect** [6]. This imperfection stems from multiple factors, including the scarcity of high-quality human-labeled data and the difficulty in formalizing high-level alignment goals into quantifiable metrics [7, 8]. For instance, consider AI alignment strategies that aim to promote safety.

---

[1]BoN and RLHF optimize an explicitly learned reward model, whereas DPO optimizes an implicit reward based on the model's log-probabilities.

39th Conference on Neural Information Processing Systems (NeurIPS 2025).

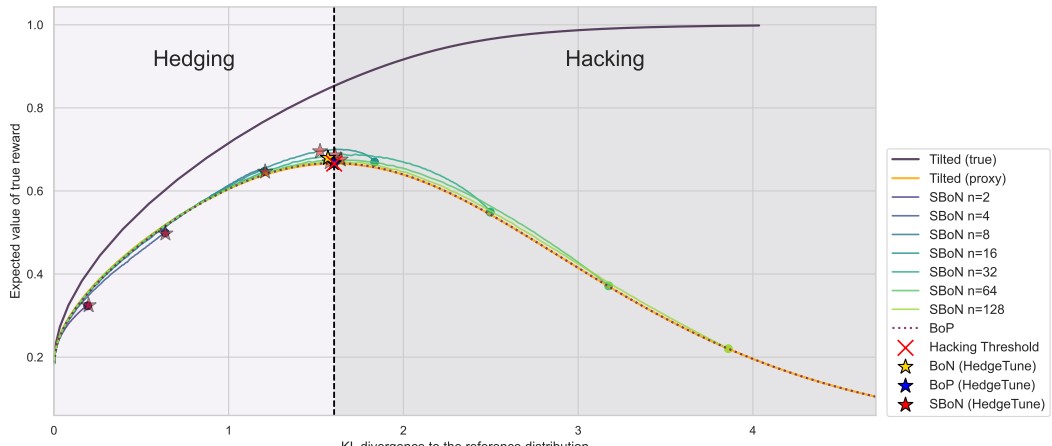

Figure 1: The mismatch between the proxy and true rewards manifests through the winner's curse. In an ideal world where we could optimize directly on the true reward, its value would rise monotonically. However, since we are optimizing for a proxy, the true reward peaks and then collapses. The point at which we find the optimal tradeoff between maximizing reward and minimizing KL divergence from the reference distribution corresponds to the *hacking threshold*. `HedgeTune` successfully recovers the hacking threshold for three inference-time mechanisms: BoN, SBoN, and BoP. In the case of BoN and BoP, `HedgeTune` recovers the optimal number of samples $n$. As for SBoN, we fix $n$ and find the corresponding inverse-temperature $\lambda$ that maximizes the true reward. If the threshold is not achievable with any $\lambda$, `HedgeTune` returns the best attainable reward, as shown for low values of $n$.

It is difficult for a single reward model to capture nuanced human user preferences and assign accurate scalar rewards in complex, context-dependent settings where safety specifications conflict [9].

In this work, we analyze and mitigate the impact of misspecified proxy rewards in inference-time alignment. Inference-time alignment has emerged as an effective and computationally efficient paradigm to improve the capabilities of large language models and align them with desired goals [10]. Among these methods, Best-of-$n$ (BoN) sampling stands out due to its simplicity and effectiveness. It generates $n$ candidate responses, scores them using a reward model, and selects the one with the highest score. Empirically, BoN demonstrates competitive performance, often matching more resource-intensive fine-tuning approaches such as RLHF and DPO [11, 12]. Additionally, BoN has received an extensive theoretical treatment [13, 14]. BoN can be asymptotically equivalent to RLHF [15], enjoys non-asymptotic guarantees [16, 17], and achieves a near-optimal winrate subject to a KL divergence constraint [18]. Methods like BoN, which generate multiple samples and choose the highest-scoring one, are victims of the **winner's curse** [19, 20, 21, 22, 23]. This phenomenon, originally identified in auction theory, occurs when selection processes systematically favor overestimates. In auctions, after each bidder submits an estimate of an item's value, the highest bid typically overestimates the true worth, causing the winner to overpay. We demonstrate the same pattern with Best-of-$n$ in Figure 1. As the number of candidates grows, the chance of picking outputs where the proxy reward overestimates true quality increases. This mismatch creates a critical tension: while initially optimizing for a proxy reward improves alignment with true goals, excessive optimization eventually leads to **reward hacking**, where the model exploits the proxy's limitations, leading to worse true performance [6, 24, 25, 26, 27]. This phenomenon is also known as Goodhart's law [28] or goal misgeneralization[2]. Such misalignment can severely degrade trust and utility, particularly in high-stakes applications [31, 32, 33].

We mathematically characterize *inference-time reward hacking* (see Theorem 1) and provide a general framework to mitigate it (see Section 4). While this phenomenon has been observed empirically in prior work [11, 14], there has been limited theoretical analysis specific to inference-time methods and ways to mitigate it; see Section 6 for a discussion. As a result, reward hacking for inference-time alignment remains a central challenge in AI alignment. The driving question behind our work is:

**When and how can we leverage useful signals from proxy rewards while mitigating hacking?**

---

[2]See [29, 30] and Section 6 for a discussion of the terminology.

We focus on answering this question for inference-time alignment methods that sample multiple responses from an LLM and use reward signals to select outputs. We develop principled *hedging techniques against the winner's curse* during inference time that precisely determine until when and how one may leverage proxy signals while preventing overoptimization.

**Overview of main results.**   Our starting point is an optimization formulation at the heart of most alignment methods: finding a distribution $\pi^*$ that maximizes a (proxy) reward $r_p$ while remaining close (in KL-divergence) to a reference $\pi_{\text{ref}}$. This is described as the following regularized optimization problem:

$$\pi^* = \underset{\pi_x \in \Delta_{\mathcal{X}}}{\operatorname{argmax}} \ \mathbb{E}_{\pi_x}\left[r_p(X)\right] - \frac{1}{\lambda} \ D_{\mathsf{KL}}(\pi_x \| \pi_{\text{ref}}) \tag{1}$$

Consider the information-theoretic regime where all distributions are known exactly. The solution of the above objective (1) is the exponential tilt of the reference distribution using the proxy reward [34]. Though theoretically interesting, tilted distributions cannot be sampled from directly; see Section 3 for a further discussion. Some attempts have been made to approximate this solution at inference time, i.e., when only samples from $\pi_{\text{ref}}$ and black-box access to $r_p$ are available. A notable example is Soft Best-of-$n$ (SBoN) [17]. In this work, we show that SBoN is an effective strategy for hedging against reward hacking due to its temperature parameter $\lambda$, which allows us to smoothly interpolate between exploiting the reward (as in BoN) and staying close to the reference. However, this comes at the expense of having two tunable parameters $(n, \lambda)$, which can be difficult to set in practice.

This motivates us to propose a new inference-time alignment strategy called **Best-of-Poisson (BoP)**. The idea behind BoP is simple: we run BoN with the number of samples $n$ chosen according to a Poisson distribution. This randomization strategy induces an exponential structure (see Eq. (2)) that closely approximates the optimal reward-tilted distribution from Eq. (1). We show that BoP achieves a nearly optimal reward-distortion tradeoff at inference. Using a single tunable parameter, BoP can approximate the optimal proxy reward-tilted solution with a KL gap of order $10^{-4}$ when rewards are uniformly distributed, allowing us to span the entire reward-distortion region at inference (see Figure 2 and Theorem 5 in Appendix C). BoP can serve as a computationally efficient stand-in for the optimal tilted distribution with negligible loss in KL-reward tradeoff.

In practice, hedging translates to selecting parameters of inference-time alignment methods to avoid overoptimization on a proxy reward. To do so, we introduce `HedgeTune`: an algorithm for tuning parameters in BoN, SBoN, and BoP in order to hedge against hacking (see Algorithm 4). We illustrate the benefit of hedging in Figure 1, where we plot the expected value of the true reward versus the distortion with respect to the reference distribution for various inference-time alignment methods. If we had access to the true reward, the optimal solution would be the tilting of the reference distribution via the true reward, leading to the reward-distortion Pareto frontier (purple curve in Figure 1). However, as we are tilting via the proxy reward, we suffer from the winner's curse: the true reward (orange curve in Figure 1) increases at first and then collapses. This behavior also manifests in BoN, as seen in the dotted points. Hedging allows us to find the *hacking threshold*: the parameters of inference-time alignment methods that yield the best tradeoff between (true) reward and distortion relative to the base model.

Our contributions are as follows:

- We mathematically formalize inference-time reward hacking (Definition 1) and derive conditions when overoptimizing imperfect proxy rewards inevitably leads to performance degradation (Theorem 1, Corollary 3).

- We introduce Best-of-Poisson (BoP), a novel inference-time alignment method (Algorithm 3). For uniformly distributed rewards, BoP approximates the optimal tilted distribution with negligible KL divergence gap (Theorem 5).

- We develop `HedgeTune`, a principled hedging framework that mitigates reward hacking by finding the optimal inference-time parameters (Algorithm 4). We empirically demonstrate that hedging strategies significantly outperform standard BoN sampling with minimal computational overhead on math, reasoning, and human-preference setups (Section 5).

---
**Algorithm 1 Best-of-$n$ Sampling (BoN)**

---
1: **Input:** Integer $n \geq 1$, base policy $\pi_{\text{ref}}$
2: Draw $n$ samples $X_1, \ldots, X_n$ i.i.d. from $\pi_{\text{ref}}$
3: Compute proxy rewards $R_i = r_p(X_i)$ for all $i$
4: Select $j = \arg\max_{i \in \{1, \ldots, n\}} r_p(X_i)$
5: **Return:** $Y = X_j$

---

## 2 Inference-Time Reward Hacking

In this section, we formalize inference-time reward hacking and show its inevitability under methods like BoN.

**Notation and Technical Assumptions.** Let $\mathcal{X}$ be a finite alphabet of tokens and let $\pi$ be a probability mass function (PMF) over token sequences $x \in \mathcal{X}^*$. We denote the probability simplex over all finite sequences of tokens as $\Delta_{\mathcal{X}^*}$. Let $\pi_{\text{ref}} \in \Delta_{\mathcal{X}^*}$ be the frozen reference policy, typically a supervised fine-tuned (SFT) model. Let $r_p : \mathcal{X}^* \to \mathbb{R}$ be a proxy reward function that assigns a unique scalar value to each sequence. This is the reward we use to optimize $\pi_{\text{ref}}$ during inference-time alignment. An inference-time alignment method parameterized by $\theta$ transforms the base policy $\pi_{\text{ref}}$ into a new distribution $\pi_\theta$. Here, $\theta$ **is the parameter of the alignment method itself** (e.g., the number of candidates $n$ in Best-of-$n$) and not of the reference policy. The performance of the aligned distribution is measured using a true reward $r_t$. We denote our measure of interest as $f(\theta) = \mathbb{E}_{U \sim \pi_\theta}[r_t(U)]$.

For theoretical tractability, we adopt the assumption that proxy rewards can be transformed to have a continuous uniform distribution, stated next. While the policy $\pi_{\text{ref}}$ itself can be complex and non-uniform, we show that we only need to consider the one-dimensional uniform distribution of the proxy reward when analyzing BoN and its variants. We later relax this assumption in the appendix and show that our results extend to the discrete case (see B.2).

**Assumption 1** (Uniform Reward Mapping). The proxy rewards $r_p(x)$ obtained by sampling sequences from the reference policy, $x \sim \pi_{\text{ref}}$, are uniformly distributed over $[0, 1]$.

Assuming uniformly distributed proxy rewards incurs little loss of generality. The proxy reward scores do not need to be uniform themselves. We first map the proxy reward scores to a standardized discrete space by transforming them into their quantiles using their own empirical CDF $F_p$ [13, 35]. This process, defined by the relation $u = F_p(r_p)$, ensures that the new reward $u$ is uniform on $[0, 1]$ by construction. This transformation preserves the rank-ordering of the scores: an output with a higher proxy reward will also have a higher transformed score. However, the resulting random variable will not be a continuous uniform random variable (which we assume), and instead will be a discrete uniform random variable. [18] has shown for a sufficiently dense set of examples, that the error incurred by this continuous model is negligible in the context of LLMs.

**Inference-time alignment.** The core challenge we address is the mismatch between the proxy reward $r_p$ and the true reward $r_t$. We focus on the family of inference-time methods that first sample a pool of candidate outputs and then use their proxy reward scores to define the selection mechanism. Two examples from this family are:

1. **Best-of-$n$** (see Algorithm 1). BoN places all probability mass on the sample with the highest proxy reward.

2. **Soft Best-of-$n$** (see Algorithm 2). SBoN is a generalization of BoN recently proposed by [17]. It applies a temperature-scaled softmax over candidate scores. As $\lambda \to 0$, SBoN sampling approaches uniform selection among the $n$ candidates. As $\lambda \to \infty$, SBoN converges to standard BoN sampling.

We first formalize inference-time reward hacking through the following definition.

**Definition 1** (Inference-Time Reward Hacking). Let $\pi_\theta$ be a distribution induced by an inference-time alignment method with parameter $\theta$, where we assume increasing $\theta$ increases both the expected proxy reward $\mathbb{E}_{X \sim \pi_\theta}[r_p(X)]$ and the KL-divergence $D_{\text{KL}}(\pi_\theta \| \pi_{\text{ref}})$. We say that *inference-time reward hacking* occurs when there exists a threshold $\theta^\dagger$ such that for $\theta > \theta^\dagger$, $\mathbb{E}_{X \sim \pi_{\theta^\dagger}}[r_t(X)] > \mathbb{E}_{X \sim \pi_\theta}[r_t(X)]$ (i.e., the true reward decreases), despite the proxy reward and KL-divergence continuing to increase. The largest value of $\theta^\dagger$ for which this holds is called the *hacking threshold*.

---

**Algorithm 2 Soft Best-of-$n$ Sampling (SBoN)**

---

1: **Input:** Integer $n \geq 1$, inverse temperature $\lambda > 0$, base policy $\pi_{\text{ref}}$
2: Draw $n$ samples $X_1, \ldots, X_n$ i.i.d. from $\pi_{\text{ref}}$
3: Compute proxy rewards $R_i = r_p(X_i)$ for all $i$
4: Sample index $Z \in \{1, \ldots, n\}$ with probability

$$\Pr(Z = i) = \frac{e^{\lambda\, r_p(X_i)}}{\sum_{j=1}^{n} e^{\lambda\, r_p(X_j)}}$$

5: **Return:** $Y = X_Z$

---

Definition 1 offers a concrete basis for operationalizing and measuring the winner's curse for inference time methods. The hacking threshold $\theta^\dagger$ is the ideal operating parameter for an inference-time alignment method. The following theorem establishes that, under common conditions, the shape of $f(\theta)$ is well-behaved: it either varies monotonically or reaches exactly one extremum.

**Theorem 1** (Inevitability of Reward Hacking). Let $\{\pi_\theta\}_{\theta \in \Theta \subset \mathbb{R}}$ be a family of distributions with density $p_\theta(x)$ on a common support $\mathcal{X}$ such that **(i)** $p_\theta(x)$ is strictly totally positive of order 2 (TP$_2$) in $(\theta, x)$, and **(ii)** its score function $\psi(x, \theta) := \partial_\theta \log p_\theta(x)$ is continuous in $x$ and strictly increasing in $x$ for each fixed $\theta$. For any bounded, non-negative true reward $r_t : \mathcal{X} \to [0, \infty)$ define

$$f(\theta) := \mathbb{E}_{X \sim \pi_\theta}[r_t(X)]$$

Then $f$ is either monotone in $\theta$ or possesses a single **unique** interior extremum $\theta^\dagger$.

**Corollary 1** (Inevitability of Reward Hacking for Strictly MLR densities). Let $p_\theta(x)$ be a *strictly monotone–likelihood–ratio* in $x$. If the score function $\psi(x, \theta) = \partial_\theta \log p_\theta(x)$ is strictly increasing in $x$, then Theorem 1 applies. In particular, this applies to Best-of-$n$, Best-of-Poisson (to be introduced in Section 3,) and to any canonical distribution from the exponential family with strictly monotone statistic and strictly monotone natural parameter.

Four scenarios may occur: (i) *monotonic improvement:* true reward continuously increases with optimization strength; (ii) *reward hacking:* true reward initially improves but deteriorates beyond a critical threshold; (iii) *reward grokking:* true reward initially declines but then improves beyond a critical threshold; (iv) *immediate decline:* any optimization immediately harms true performance. We describe exactly when each regime occurs for MLR densities in Corollary 3.

The conditions in Theorem 1 guarantee that the inference-time policies behave in a "well-ordered" manner: increasing the tuning parameter $\theta$ consistently makes the policy "greedier" and more likely to select outputs with high proxy rewards. The clearest example is the $n$ in Best-of-$n$: the larger the $n$, the more aggressively we optimize for the proxy. The unimodality of the true reward function then renders the problem of locating the optimal operating point $\theta^\dagger$ algorithmically tractable. To implement this insight, we develop hedging strategies that balance the exploitation of the proxy reward against the fidelity to the reference distribution. Each inference-time method offers a parameter controlling this proxy reward-KL tradeoff: $n$ in BoN, $\lambda$ in SBoN (for a fixed $n$), and $\mu$ in BoP (introduced in Section 3). Before introducing methods for tuning inference-time alignment methods, we first introduce **Best-of-Poisson** sampling: an alternative to BoN that approximates the optimal tilted distribution.

## 3 Best-of-Poisson: Approximating the Optimal Reward

While Soft Best-of-$n$ offers a principled approach to mitigate reward hacking, it requires tuning both the number of samples $n$ and the temperature parameter $\lambda$. In this section, we introduce Best-of-Poisson (BoP) (Algorithm 3), that is provably close to the solution of (1) with a single tunable parameter (Figure 2). BoP is of independent interest as it provides a mathematically elegant and computationally efficient way to near-optimally span the entire reward-KL distortion region with a single parameter. The key insight behind BoP is to replace the fixed sample size $n$ in BoN with a random sample size drawn from a Poisson distribution.

The parameter $\mu$ in BoP controls the expected number of samples, analogous to how $n$ functions in BoN. We first sample $n'$ from a Poisson distribution parameterized by $\mu$ and set $n = n' + 1$ to ensure

---

**Algorithm 3 Best-of-Poisson Sampling (BoP)**

---

1: **Input:** Poisson parameter $\mu > 0$, base policy $\pi_{\text{ref}}$
2: Sample $n' \sim \text{Poisson}(\mu)$ and set $n = n' + 1$
3: Draw $n$ samples $X_1, \ldots, X_n$ i.i.d. from $\pi_{\text{ref}}$
4: Compute proxy rewards $R_i = r_p(X_i)$ for all $i$
5: Select $j = \arg\max_{i \in \{1,\ldots,n\}} r_p(X_i)$
6: **Return:** $Y = X_j$

---

at least one sample is generated. Under Assumption 1, the BoP distribution with parameter $\mu$ has a probability density function given by (see Appendix C)

$$q_\mu(x) = (\mu x + 1)e^{\mu(x-1)} \text{ for } x \in [0, 1]. \tag{2}$$

The following theorem characterizes the KL divergence and expected value of BoP:

**Theorem 2** (KL Divergence and Expected Value of BoP). Let $X_{\text{BoP}}$ be the random variable representing the response selected by BoP with parameter $\mu$. Then:

$$\text{KL}(\pi_{\text{BoP}} \| \pi_{\text{ref}}) = \frac{e^{-\mu-1}(\text{Ei}(\mu + 1) - \text{Ei}(1))}{\mu} + \log(\mu + 1) - 1. \tag{3}$$

$$\mathbb{E}[X_{\text{BoP}}] = 1 - \frac{1}{\mu} + \frac{1 - e^{-\mu}}{\mu^2}. \tag{4}$$

where $\text{Ei}(z) = -\int_{-z}^{\infty} \frac{e^{-t}}{t} dt$ is the exponential integral function.

What makes BoP particularly valuable is its ability to closely approximate the solution of (1), i.e., the optimal KL-constrained tilted distribution with parameter $\lambda > 0$, defined as $\pi_\lambda^*(x) = \frac{\pi_{\text{ref}}(x)e^{\lambda r_p(x)}}{Z(\lambda)}$, where $Z(\lambda)$ is the normalization constant. While this distribution is theoretically optimal for balancing reward and divergence, computing it is intractable. To draw a next token from the tilted $\pi_\lambda^*$ for an autoregressive LLM, one would have to compute

$$\pi_\lambda^*(x_{t+1} \mid x_{\leq t}) = \frac{\pi_{\text{ref}}(x_{\leq t+1}) \, e^{\lambda r_p(x_{\leq t+1})}}{\sum_{x'} \pi_{\text{ref}}(x_{\leq t}x') \, e^{\lambda r_p(x_{\leq t}x')}},$$

where the denominator sums over every possible continuation of the prefix $x_{\leq t}$. Because the space of continuations grows exponentially with the remaining sequence length, evaluating this denominator (and hence sampling a single token) is computationally prohibitive for LLMs.

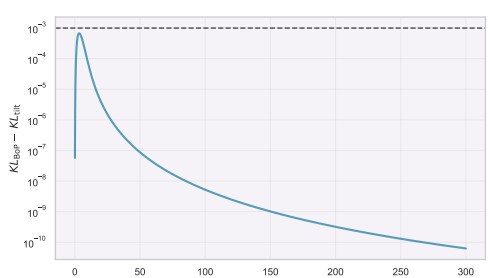

Figure 2: The difference in KL divergence when BoP and optimal tilted distributions are matched to produce the same expected reward. The extremely small gap (of order $10^{-4}$) demonstrates that BoP approximates the optimal distribution with negligible performance loss

Our analysis (Appendix C) shows that BoP provably achieves nearly identical performance to this optimal distribution with minimal KL divergence gap. In Appendix C.3, we also introduce a natural extension of Soft Best-of-$n$ and Best-of-Poisson, which we call *Soft Best-of-Poisson (SBoP)*. As illustrated in Figure 2, the KL divergence gap between these distributions is remarkably small. Numerical evaluation confirms that for all $\mu$, if $\lambda > 0$ is chosen so that $\mathbb{E}_{X \sim \pi_{\text{BoP}}}[r_p(X)] = \mathbb{E}_{X \sim \pi_\lambda^*}[r_p(X)]$, then the KL-gap is bounded between 0 and $8 \times 10^{-4}$.

This near-equivalence means that BoP can serve as a practical stand-in for the theoretically optimal tilted distribution. This result has two consequences. First, hedging in the optimal tilted distribution is almost equivalent to hedging with BoP. Second, Equation (1) represents the solution to the standard RLHF optimization problem, which implies that BoP is an inference-time approximation to RLHF and allows us to use BoP to easily traverse between policies rather than having to finetune a new model for each $\lambda$ of interest. In the next section, we turn to the question of how to choose the right parameter value to avoid reward hacking.

**Algorithm 4** `HedgeTune`: Parameter Optimization for Hedging

---

1: **Inputs:** Proxy and true rewards $\{s_{t,k}, r_{t,k}\}$ per prompt $t$; parameter domain $\Theta$
2: **Output:** Optimal hedge parameter $\theta^{\star}$

3: STEP 1. For each prompt $t$, sort responses by their proxy scores and map their ranks to empirical quantiles $u_{t,k} \in (0,1)$.

4: STEP 2. Specify the score function $\psi(u, \theta)$ and density $p_\theta(u)$ according to the inference-time method (e.g., BoN, SBoN, BoP; see Appendix D).

5: STEP 3. For a given $t$ and $\theta \in \Theta$, define the residual $R_t(\theta) = \mathbb{E}_{u \sim p_\theta}[r_t(u)\,\psi(u, \theta)]$. This can be estimated from the empirical pairs $\{(u_{t,k}, r_t(u_{t,k}))\}$.

6: STEP 4. Find $\theta^{\star} \in \Theta$ such that the average residual $\bar{R}(\theta^{\star}) = \frac{1}{|T|}\sum_t \hat{R}_t(\theta) = 0$ via one-dimensional root-finding.

---

# 4 Hedging to mitigate reward hacking

In this section, we develop a unified framework for choosing the inference-time parameter $\theta$ in order to maximize the expected true reward and avoid hacking. The main limitation is that we require black-box access to the true reward to perform a *one-time calibration* of the parameter $\theta$. This is practical in several common scenarios. One example is *domains with verifiable ground truth*, such as mathematical reasoning, program synthesis, or factual question answering. One may also opt to use an LLM-as-a-judge or a more powerful but computationally expensive reward model [36, 37].

Assume that we are given the proxy and true reward scores for a set of query-response pairs. By first constructing an empirical CDF over the generated proxy reward scores, we transform these proxy scores to have a uniform distribution. We denote the transformed proxy reward as $U$. Each sampling method (BoN, SBoN, and BoP) induces a distribution $\pi_\theta$ over proxy-percentiles $u \in [0,1]$, where $\theta$ is the corresponding parameter (sample size $n$, inverse-temperature $\lambda$, or Poisson rate $\mu$). Since we know by Theorem 1 that the expected true reward has at most one peak, our key insight is to create the precise **hedge** against hacking by finding the parameter value where the marginal benefit of increasing the proxy reward equals zero. We present the following conditions that the hacking threshold must satisfy for each of the three inference-time methods.

**Theorem 3** (Hacking Threshold Characterization). Let $r_t$ be a true reward oracle and $\theta^\dagger$ be the hacking threshold from Definition 1. For each inference-time method, $\theta^\dagger$ is characterized by the following conditions:

For BoN, $n^\dagger$ satisfies:

$$\nabla_\alpha \mathbb{E}_{u \sim \mathrm{Beta}(\alpha, 1)}[r_t(u)] = \int_0^1 r_t(u)\left(\frac{1}{n^\dagger} + \ln u\right) u^{n^\dagger - 1}\, du = 0. \tag{5}$$

For SBoN, $\lambda^\dagger$ satisfies:

$$\nabla_\lambda \mathbb{E}_{u \sim f_\lambda}[r_t(u)] = \mathrm{Cov}_{u \sim f_{\lambda^\dagger}}(r_t(u), u) = 0, \tag{6}$$

For BoP, $\mu^\dagger$ satisfies:

$$\nabla_\mu \mathbb{E}_{u \sim f_\mu}[r_t(u)] = \mathbb{E}_{u \sim f_{\mu^\dagger}}\left[r_t(u)\left(u - 1 + \frac{u}{\mu^\dagger u + 1}\right)\right] = 0. \tag{7}$$

The proof can be found in Appendix D. Consequently, we provide `HedgeTune` (Algorithm 4), an algorithm that numerically solves the corresponding root-finding problem to determine the optimal inference-time parameter for BoN, SBoN, or BoP. **Note that we do not need access to the LLM distribution itself**. We use the explicit expressions of the *score* function $\psi$ and estimate the residual function $R(\theta) = \mathbb{E}[r_t(u)\psi(u, \theta)]$ which captures the alignment between the true reward and the proxy-weighted score. The optimal parameter $\theta^\dagger$ is found efficiently as the root of this function using standard methods such as bisection or Newton's method [38] and can later be used directly at inference.

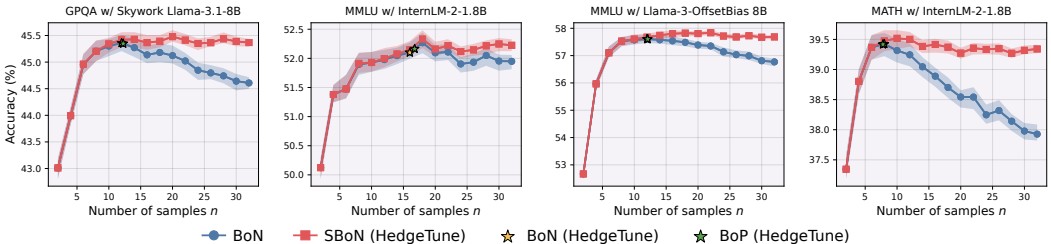

Figure 3: **Hedging mitigates hacking in verifiable reward setups.** We plot the expected accuracy on various benchmarks versus the number of samples $n$. HedgeTune successfully recovers the best operating point for BoN and BoP and provides a superior reward-distortion curve with SBoN.

## 5 Experiments: Hedging in Practice

In this section, we validate that hedging is an effective tool against reward hacking and can provide superior reward-distortion tradeoffs in two experimental setups: one with verifiable rewards and one with human-preference data.

### 5.1 Hacking in verifiable setups.

We first demonstrate that hedging mitigates reward hacking on standard verifiable benchmarks such as MMLU Pro and GPQA. Specifically, we use the open-source *Preference Proxy Evaluations* (**PPE**) dataset [39], which contains multiple responses per benchmark question from frontier models such as GPT-4o-mini and Claude Haiku 3. Each response is then scored by a suite of reward models. We focus on the benchmark dataset–reward model pairs identified in **PPE** as exhibiting reward hacking under Best-of-$n$ sampling. We then apply HedgeTune to find the optimal operating point for BoN, BoP, and SBoN.

**Reward Models.** We consider three reward models of varying sizes: InternLM-2 1.8B [40], Llama-3-Offset-Bias 8B [41], and Skywork-Llama-3.1 8B [42].

**Datasets.** We consider three datasets: MMLU Pro (complex reasoning) [43], MATH (mathematical problem solving) [44], and GPQA (questions in natural sciences) [45]. A correct response gets a true reward of 1, while an incorrect response gets a true reward of 0.

**Findings.** The observed reward hacking curve in Figure 3 matches our theoretical prediction in Theorem 1. For example, BoN demonstrates hacking on the GPQA dataset, even with the very capable Skywork-Llama 3.1 8B as a proxy reward (currently ranked the 12th best non-generative reward model on RewardBench [37]). HedgeTune recovers the best operating points in all the cases.

### 5.2 Hacking in the wild with human preferences.

Our experimental design follows the methodology of Coste et al. [46] and Gao et al. [11], wherein proxy reward models are trained using preferences of a fixed gold reward model. In many real-world cases, we do not have access to the gold reward and instead have access to preference data. However, as we demonstrate below, a favorable operating point can still be found using traditional hyperparameter search.

**Models.** As a reference model, we use a 1.4B Pythia model [47] fine-tuned on AlpacaFarm dataset, but without any subsequent alignment (e.g., RLHF or DPO). This reference model is used to generate responses. We use **AlpacaRM** [48] as our gold reward model. **AlpacaRM** is an established reward model trained on human preference data and has been adopted in prior work on reward model evaluation [46, 49, 50]. This model serves as the ground truth for generating preference labels for training proxy reward models. As for proxy rewards, we use the setup of Coste et al. [46] where we train Pythia 44m models.

**Datasets.** We use the `tlc4418/gold_labelled_gens` dataset from Coste et al. [46]. This dataset comprises of 12,600 responses generated by the Pythia 1.4B base policy for each of 1,000 prompts. The prompts are sourced from the validation split of the AlpacaFarm dataset [48]. Each generated

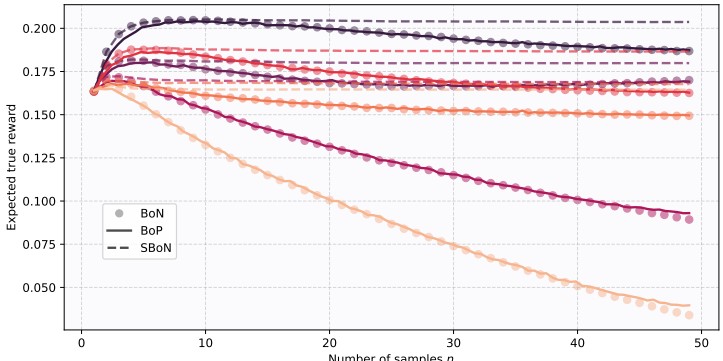

Figure 4: **Hedging mitigates hacking in human-preference setups.** We use three inference-time methods (BoN, SBoN, and BoP) on trained proxy rewards. Hacking is effectively mitigated by hedging via $\lambda$ in SBoN or $n$ in BoN and BoP.

response in this dataset is scored by **AlpacaRM**. To train proxy reward models, we construct preference datasets using the **AlpacaRM** reward scores. For each training instance, we sample a pair of responses to a given prompt and label them based on their given true reward scores.

**Training.** The proxy RMs are trained using a standard binary cross-entropy loss on preference pairs. We train proxy RMs on preference pair datasets of varying sizes: 10k, 20k, 46k, and 80k. In line with [46, 51, 52], we simulate disagreements in human annotators by considering two cases: (a) no label noise in the preferences, and (b) 25% label noise. All proxy RM training runs are repeated across 4 random seeds each. We present some of the runs with 25% label noise in Figure 4 and we present the remaining results in Appendix E, along with other hyperparameters and training details. Post training, we use each proxy model to score a set of 800 prompts, with 12,600 responses each.

**Findings.** We apply BoN, SBoN, and BoP on each run and find the expected value of the true reward as a function of $n$. When reward hacking manifests, we find a hacking threshold for BoN and BoP that maximizes their reward. For SBoN, with a selected $\lambda^{\dagger}$, we attain the peak value without suffering from reward hacking. Meanwhile, if the proxy is always at odds with the true reward, the optimal solution is the reference distribution itself, corresponding to $\lambda = 0$.

## 6    Related Work

**Reward Hacking.** Reward hacking has been widely studied in RL literature [8, 7, 53], also under the name misspecification [30], goal misgeneralization [54], or specification gaming [55].

In the context of LLMs, overoptimization has been referred to as reward hacking or Goodhart's Law [28, 11, 26, 27]. Hacking behavior has been found to manifest in unwanted or surprising behavior [56, 57] across a variety of tasks [58, 11, 14]. Prior works have proposed various formulations of reward hacking based on true performance behavior [24], correlation between proxy and true reward [6], or distribution shift [25]. [14] prove that BoN alignment provably suffers from reward hacking when the number of samples $n$ is large.

To address inference time hacking, a variety of methods have been explored to varying success, such as ensembling [46, 59, 60, 61], regularization [62] or rejection sampling [14]. [63] simulate $n$ particles resampled using a softmax reward to improve performance of reasoning models, similar to SBoN over reasoning steps. However, all methods suffer from some combination of additional generation cost beyond generating $n$ samples and estimation of additional side-quantities such as KL-divergence or $\chi^2$-divergence. Additionally, a variety of approaches have been proposed to mitigate reward hacking during RLHF finetuning such as regularization [64, 52, 65, 66], $\chi^2$-divergence [67, 6], uncertainty estimation [68], and reward pessimism [69, 70] although [26] demonstrated that RLHF can still result in reward hacking under heavy-tailed reward mismatch. Finally, prior works have also focused on improving reward models to prevent mismatch and reduce hacking [71, 72, 73, 74, 51, 75].

**Best-of-$n$.** Best-of-$n$ sampling is a simple inference-time approach for alignment [1, 2, 76]. Prior results have characterized the expected reward gap and KL divergence between BoN sampling and the reference model and have demonstrated that BoN is asymptotically equivalent to KL-constrained reinforcement learning [13, 15, 16]. There have been various methodological improvements on BoN sampling. One such improvement is to reduce the cost of sampling $n$ sequences via tree-based or speculative search [77, 78] . Additionally, [18, 79, 80, 81, 82] distill the BoN sampling distribution into a model via fine-tuning. Finally, [35, 83] propose inference-aware methods to improve BoN. Other works focus on improving the reward model through self-training [84]. In this work, we focus on a variant of BoN, Soft Best-of-$n$ [17], which allows for finer control between sampling from the base model and the reward-maximizing generation.

## 7    Conclusion

Our work tackles the fundamental challenge that all proxy rewards are imperfect, yet they remain essential for guiding and improving AI systems. We establish a theoretical framework proving the inevitability of reward hacking in inference-time alignment and introduce practical hedging strategies to mitigate its harmful effects. By developing *Best-of-Poisson* sampling which achieves near-optimal reward-distortion tradeoffs with a single parameter and the `HedgeTune` algorithm for precisely calibrating inference methods, we enable practitioners to extract valuable signals from proxy rewards without falling prey to Goodhart's law.

We also emphasize that AI safety concerns are, at their heart, socio-technical [30]. The impact of model failures due to hacking (and beyond) will depend on the application and stakeholders at hand. One clear example is an AI agent that generates controversial and toxic content on social media to optimize its engagement metric at the direct expense of content quality and safety [85]. Other dangerous failure patterns include models that learn to be sycophantic instead of truthful [86] or actively exploit loopholes to circumvent oversight [32]. By studying the phenomenon of hacking from a rigorous mathematical perspective, our work helps with the design of hacking mitigation (and hence AI safety methods) with provable performance guarantees. Technical approaches such as ours are important for AI safety in practice, although we recognize that they must exist within a larger safety ecosystem where rigorous technical foundations inform responsible deployment practices and policy decisions, and vice-versa. Ultimately, this work demonstrates that principled hedging is a promising direction for building safer, more reliable AI systems.

## Acknowledgments and Disclosure of Funding

We thank the anonymous reviewers for their helpful comments and suggestions. This work is supported by the National Science Foundation under grants CIF 2312667, FAI 2040880, and CIF 2231707. This work is supported in part by NSF Awards IIS-2008461, IIS-2040989, and IIS-2238714; an AI2050 Early Career Fellowship from Schmidt Sciences; and research awards from Google, OpenAI, the Harvard Data Science Initiative, and the Digital, Data, and Design (D^3) Institute at Harvard. AO is supported by the National Science Foundation Graduate Research Fellowship under Grant No. DGE-2140743. The views expressed are those of the authors and do not necessarily reflect the official policies or positions of the funding organizations.

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

# Inference-Time Reward Hacking in
# Large Language Models

## A  Overview

In this supplementary material, we provide the following:

- Appendix B provides additional proofs and results on the inevitability of reward hacking.
- Appendix C provides additional proofs for Best-of-Poisson, as well as Soft Best-of-Poisson.
- Appendix D discusses `HedgeTune` in more detail.
- Appendix E provides additional details on our experimental setup.

## B  Inference-Time Reward Hacking

Increasing optimization pressure on a proxy objective can initially improve true performance, but beyond a certain point— which we call the *hacking threshold*—further optimization can lead to misalignment and degrade performance. Theorem 1 in Section 2 formalizes this phenomenon: under very general conditions on a one-parameter family of proxy distributions $\pi_\theta$, the map

$$\theta \;\mapsto\; \mathbb{E}_{X \sim \pi_\theta}[r_{\mathrm{t}}(X)] := f(\theta) \tag{8}$$

can have at most one interior extremum. Thus, there is either a monotonic benefit (or disadvantage) to strengthening the proxy or exactly one "sweet spot" before reward hacking sets in. The rest of this section proceeds as follows. We prove the single-crossing property for the derivative of the true reward by invoking variation-diminishing kernels, and then we instantiate our results with Monotone-Likelihood-Ratio densities in Corollaries 2 and 3. We specialize the discussion to two concrete examples with Best-of-$n$ and Best-of-Poisson. Next, we extend our result to the more general discrete case in B.2.

### B.1  Uniform Case

The following results hold under our Assumption 1. Let $\psi(x, \theta)$ denote the score function of distribution $\pi_\theta(x)$ with density $p_\theta(x)$. Standard calculations under mild regularity conditions gives us the derivative of the true reward under $\pi_\theta$:

$$f'(\theta) = \int r_t(x)\nabla_\theta p_\theta(x)\mathrm{d}x = \int r_t(x)p_\theta(x)\nabla_\theta \log p_\theta(x)\mathrm{d}x = \mathbb{E}_{X \sim \pi_\theta}[r_t(X)\psi(X, \theta)] \tag{9}$$

The main idea to establish Theorem 1 is that when we use parameter $\theta$ to control inference-time methods, we create a family of densities $\{p_\theta(x)\}$ that act as positive kernels. These kernels satisfy the strict total positivity conditions required for variation-diminishing theorems to apply. Our only assumption on the true reward is that it is bounded. We then translate the reward function so that it is non-negative. The boundedness assumption is a natural one in the alignment setting because real-world rewards originate from human judgments given on finite scales (e.g. star ratings, Likert scores, or normalized preference probabilities). Moreover, clipping or normalizing the reward prevents unbounded returns, improving the stability of policy updates and inference-time mechanisms. .

*Proof.* Fix $\theta$ and set $h_\theta(x) := r_t(x)\psi(x, \theta)$. Because $\psi(\cdot, \theta)$ is strictly increasing, it has at most one zero. Since $r_t \geq 0$, $h_\theta$ has the same single sign change in $x$. Strict $\mathrm{TP}_2$ of $p_\theta$ and Karlin's variation–diminishing theorem imply that $\theta \mapsto F(\theta) := \int h_\theta(x)p_\theta(x)\,dx = f'(\theta)$ inherits *at most* the same number of sign changes, namely one. $\qquad\square$

Having established the inevitability result, we next explore how it specializes in classical families via a simple corollary.

**Corollary 2** (Strict MLR densities). Let $p_\theta(x)$ be *strictly monotone–likelihood–ratio* in $x$ (i.e. $p_{\theta_2}(x)/p_{\theta_1}(x)$ strictly increases in $x$ whenever $\theta_2 > \theta_1$). If $\psi(x, \theta) = \partial_\theta \log p_\theta(x)$ is strictly increasing in $x$, then all conclusions of Theorem 1 apply. In particular, this applies for any regular canonical exponential family with strictly monotone statistic $T$ and strictly monotone natural parameter.

The conditions are satisfied by two inference-time methods we study:

**Best-of-$n$**: The distribution corresponds to the maximum of $n$ i.i.d. samples from the reference distribution. When proxy rewards are uniformly distributed, this yields $p_n(u) = nu^{n-1}$ for $u \in [0, 1]$. The likelihood ratio $\frac{p_{n_2}(u)}{p_{n_1}(u)} = \frac{n_2}{n_1} u^{n_2-n_1}$ is strictly increasing in $u$ when $n_2 > n_1$, establishing strict MLR (which implies strict TP2). The score function $\psi(u, n) = \frac{1}{n} + \log u$ is strictly increasing in $u$.

**Best-of-Poisson**: The distribution $p_\mu(u) = (\mu u + 1)e^{\mu(u-1)}$ for $u \in [0, 1]$ can be verified to satisfy strict MLR by direct computation of likelihood ratios. The score function $\psi(u, \mu) = u - 1 + \frac{u}{\mu u + 1}$ is strictly increasing in $u$.

Under the conditions presented in Theorem 1, we know that at most one interior extrema exists. The following corollary gives precise conditions on when such an interior extrema exists.

**Lemma 1.** Under the assumptions of Theorem 1 and with $r_t \in C^1$ on a neighborhood of its boundaries (0 and 1) and let $\Theta = [\theta_l, \theta_r]$,

$$\lim_{\theta \downarrow \theta_\ell} f'(\theta) = r_t'(0+) \, \mathbb{E}_{\theta_\ell}[X \, \psi(X, \theta_\ell)], \qquad \lim_{\theta \uparrow \theta_r} f'(\theta) = - r_t'(1-) \, \mathbb{E}_{\theta_r}[(1 - X)\psi(X, \theta_r)].$$

Then, a stationary point exists *iff*

$$\lim_{\theta \downarrow \theta_\ell} f'(\theta) \quad \text{and} \quad \lim_{\theta \uparrow \theta_r} f'(\theta)$$

are of opposite sign (or one limit is 0 while the other is non-zero).

*Proof.* We prove the left boundary results (the right one is identical with $x \mapsto 1 - x$.) Write the first-order expansion $r_t(x) = r_t(0+) + r_t'(0+)x + R(x)$ with $R(x) = o(x)$ as $x \to 0$. Because $\mathbb{E}_\theta[\psi] = 0$,

$$f'(\theta) = r_t'(0+)\mathbb{E}_\theta[X\psi] + \mathbb{E}_\theta[R(X)\psi].$$

Strict increase of $\psi$ implies $|X\psi| \leq C(1 + X)$ on $[0, 1] \times [\theta_\ell, \theta_\ell + \rho]$, so $\mathbb{E}_\theta[X\psi] \to \mathbb{E}_{\theta_\ell}[X\psi]$ by dominated convergence. Next, fix $\delta \in (0, 1)$ and split the expectation:

$$\mathbb{E}_\theta[r_t(X)\psi] = \mathbb{E}_\theta[r_t(X)\psi \mathbf{1}_{\{X \leq \delta\}}] + \mathbb{E}_\theta[r_t(X)\psi \mathbf{1}_{\{X > \delta\}}].$$

Near 0, we have $|R(x)| \leq c_\delta x$, hence the term is bounded by $c_\delta \mathbb{E}_\theta[X|\psi|]$; choose $\delta$ so small that $c_\delta \mathbb{E}_{\theta_\ell}[X|\psi|] < \varepsilon$ and continuity keeps it $< 2\varepsilon$ for $\theta$ close enough to $\theta_\ell$. Away from 0, we use boundedness of $r_t$ and local boundedness of $\psi$ to obtain a factor $P_\theta\{X > \delta\} \to 0$.

Combining the two parts gives $\mathbb{E}_\theta[R(X)\psi] \to 0$, yielding the claimed limit. By continuity, the Intermediate-Value Theorem then forces one root of $f'$ and single–crossing rules out a second. $\square$

**Corollary 3** (Reward behavior for Strict MLR densities). Assume Lemma 1 holds and that the family $\{p_\theta\}_{\theta \in \Theta}$ is strictly *monotone–likelihood–ratio* in $x \in (0, 1)$. Then for every $\theta \in \Theta$

$$L_\theta := \mathbb{E}_\theta[X \, \psi(X, \theta)] > 0, \qquad R_\theta := \mathbb{E}_\theta[(1 - X)\psi(X, \theta)] < 0,$$

so

$$\text{sign} f'(\theta_\ell^+) = \text{sign} \, r_t'(0+), \qquad \text{sign} f'(\theta_r^-) = \text{sign} \, r_t'(1-)$$

Single-crossing of $f'$ implies that $f(\theta)$ can assume *exactly* one of the four shapes:

| regime | $r_t'(0+)$ | $r_t'(1-)$ |
|---|---|---|
| monotonic improvement | $\geq 0$ | $\geq 0$ |
| reward hacking | $> 0$ | $< 0$ |
| reward grokking | $< 0$ | $> 0$ |
| immediate decline | $\leq 0$ | $\leq 0$ |

*Proof.* Let $p_\theta$ be differentiable in $\theta$ with score $\psi(x,\theta) = \partial_\theta \log p_\theta(x)$. For any integrable $g$ we have shown that:

$$\frac{d}{d\theta}\mathbb{E}_\theta[g(X)] = \mathbb{E}_\theta[g(X)\psi(X,\theta)]$$

We first use that the MLR property implies first-order stochastic dominance. For every *increasing* function $g$, we have that $\theta \mapsto \mathbb{E}_\theta[g(X)]$ is non-decreasing and its derivative is non-negative. Choosing $g(x) = x$ gives

$$L_\theta = \mathbb{E}_\theta[X\,\psi(X,\theta)] = \frac{d}{d\theta}\mathbb{E}_\theta[X] \; > \; 0$$

On the other hand, $g(x) = 1 - x$ (strictly decreasing) yields:

$$R_\theta = \mathbb{E}_\theta[(1 - X)\psi(X,\theta)] = \frac{d}{d\theta}\mathbb{E}_\theta[1 - X] \; < \; 0$$

Thus $L_\theta > 0$ and $R_\theta < 0$ for every $\theta \in \Theta$. For example. for BoN, $\psi(x,n) = 1/n + \log x$ with $\mathbb{E}_n[X\,\psi] = 1/(n+1)^2$, $\mathbb{E}_n[(1-X)\psi] = -1/(n+1)^2$. Hence a stationary point exists iff $r_t'(0+)$ and $r_t'(1-)$ have opposite signs, and its location $n_\star$ solves $\mathbb{E}_n[r_t(X)\psi(X,n)] = 0$. $\qquad\square$

We have shown the conditions under which the expected value of true reward has a critical point with respect to $\theta$. We now show the conditions under which our results extend identically if we are studying the expected value of true reward as a function of the KL divergence with respect to the reference distribution $\pi_{\theta_0}$.

**Lemma 2.** Let $\{\pi_\theta\}_{\theta \in \Theta}$ be a regular parametric family with density $p_\theta(x)$ such that

- $p_\theta$ and $\partial_\theta p_\theta$ are jointly measurable and $\partial_\theta p_\theta(x)$ is locally integrable in $\theta$;

- the *score* $\psi(x,\theta) := \partial_\theta \log p_\theta(x)$ is square–integrable: $\mathbb{E}_\theta[\psi^2] < \infty$.

For a fixed reference point $\theta_0 \in \Theta$ define the Kullback–Leibler divergence

$$D(\theta\|\theta_0) \; := \; \int p_\theta(x)\,\log\frac{p_\theta(x)}{p_{\theta_0}(x)}\,d\mu(x).$$

Then $D(\theta\|\theta_0)$ is differentiable and

$$\frac{d}{d\theta}\,D(\theta\|\theta_0) \; = \; \mathbb{E}_\theta\Big[(\log p_\theta(x) - \log p_{\theta_0}(x))\,\psi(X,\theta)\Big].$$

In particular, for canonical exponential families with strictly increasing natural parameter, this simplifies to

$$\frac{d}{d\theta}\,D(\theta\|\theta_0) = (\eta(\theta) - \eta(\theta_0))\,A''(\theta),$$

which is strictly positive when $\theta > \theta_0$ (and negative for $\theta < \theta_0$).

*Proof.* Write $g_\theta(x) := \log p_\theta(x) - \log p_{\theta_0}(x)$. Then $D(\theta\|\theta_0) = \mathbb{E}_\theta[g_\theta(X)]$. For a *parameter-dependent* integrand the classical Fisher–Leibniz rule gives

$$\frac{d}{d\theta}\mathbb{E}_\theta[g_\theta(X)] = \mathbb{E}_\theta[\partial_\theta g_\theta(X)] + \mathbb{E}_\theta[g_\theta(X)\,\psi(X,\theta)],$$

whenever $\partial_\theta g_\theta$ exists and an $L^1$ dominated–convergence bound holds (true here by the square–integrable score assumption). Since $\partial_\theta g_\theta(x) = \psi(x,\theta)$ and $\mathbb{E}_\theta[\psi] = 0$, the first term vanishes, leaving exactly

$$\frac{d}{d\theta}D(\theta\|\theta_0) = \mathbb{E}_\theta[g_\theta(X)\,\psi(X,\theta)]$$

$\qquad\square$

## B.2 Discrete Case

In this subsection, we show that Theorem 1 holds for the general discrete case and then verify that it holds for specific policy families like Best-of-$n$ and Best-of-Poisson.

Let $Y = \{y_1, y_2, \ldots, y_m\}$ be a finite, ordered set of possible responses, where the ordering is determined by a proxy reward function $r_p$. We represent this space by the ordered index set $X = \{1, 2, \ldots, m\}$. Let $\{\pi_\theta\}_{\theta \in \Theta}$ be a one-parameter family of policies on $X$, where $\theta \in \Theta \subset \mathbb{R}$ is a continuous tuning parameter. The probability of selecting response $i$ is given by the Probability Mass Function (PMF) $\pi_\theta(i)$. Let $r_t : X \to \mathbb{R}_{\geq 0}$ be a bounded, non-negative true reward function. Our objective is to analyze the shape of the expected true reward function:

$$f(\theta) = \mathbb{E}_{X \sim \pi_\theta}[r_t(X)] = \sum_{i=1}^{m} \pi_\theta(i) \cdot r_t(i).$$

We will prove that under general conditions on the policy family $\pi_\theta$, the function $f(\theta)$ is either monotonic or has a unique interior extremum.

**Theorem 4** (Inevitability of Reward Hacking for Discrete Policies). Let $\{\pi_\theta\}_{\theta \in \Theta}$ be a family of PMFs on $X = \{1, \ldots, m\}$. Assume that:

1. The PMF $\pi_\theta(i)$ is strictly TP$_2$ in the pair $(\theta, i)$. That is, for any $\theta_1 < \theta_2$ and $i_1 < i_2$,

$$\pi_{\theta_1}(i_1)\pi_{\theta_2}(i_2) - \pi_{\theta_1}(i_2)\pi_{\theta_2}(i_1) > 0.$$

2. The score function $\psi(i, \theta) := \frac{\partial}{\partial \theta} \log \pi_\theta(i)$ exists and is strictly increasing in $i$ for each fixed $\theta$.

Then, for any bounded, non-negative true reward function $r_t(i)$, the derivative $f'(\theta)$ changes sign at most once. Hence, $f(\theta)$ is either monotonic or has a unique interior extremum.

*Proof.* We differentiate:

$$f'(\theta) = \frac{d}{d\theta} \sum_{i=1}^{m} \pi_\theta(i) r_t(i) = \sum_{i=1}^{m} \pi_\theta(i) \psi(i, \theta) r_t(i) = \mathbb{E}_{\pi_\theta}[r_t(i)\psi(i, \theta)].$$

Define $h_\theta(i) = r_t(i)\psi(i, \theta)$. Since $r_t(i) \geq 0$ and $\psi(i, \theta)$ is strictly increasing in $i$, the product $h_\theta(i)$ has at most one sign change. Since $f'(\theta) = \sum_{i=1}^{m} \pi_\theta(i) h_\theta(i)$, and $\pi_\theta$ is strictly TP$_2$, Karlin's variation-diminishing theorem implies that $f'(\theta)$ changes sign at most once, and $f(\theta)$ has at most one extremum. $\square$

We now verify that Best-of-$n$ and Best of Poisson satisfy the assumptions. Let the base PMF be $p_i$, with CDF $F_k := \sum_{i=1}^{k} p_i$.

**Best-of-$n$:** It is shown in [13] that the BoN policy has the following PMF:

$$\pi_n(i) = F_i^n - F_{i-1}^n.$$

We first verify the TP2 condition. We consider:

$$L(i) := \frac{\pi_{n_2}(i)}{\pi_{n_1}(i)} = \frac{F_i^{n_2} - F_{i-1}^{n_2}}{F_i^{n_1} - F_{i-1}^{n_1}}.$$

By Cauchy's Mean Value Theorem (CMVT), for $g(t) = t^{n_2}$, $f(t) = t^{n_1}$, and $a = F_i, b = F_{i-1}$, we get:

$$\frac{g(F_i) - g(F_{i-1})}{f(F_i) - f(F_{i-1})} = \frac{g'(c_i)}{f'(c_i)} = \frac{n_2 c_i^{n_2-1}}{n_1 c_i^{n_1-1}} = \frac{n_2}{n_1} c_i^{n_2-n_1}$$

for some $c_i \in (F_{i-1}, F_i)$. Since $F_i$ is strictly increasing, $c_i < c_{i+1}$, and $c_i^{n_2-n_1}$ is strictly increasing. Hence $L(i+1) > L(i)$. Next, we note that the score function is:

$$\psi(i, n) = \nabla_n \log(F_i^n - F_{i-1}^n).$$

Let $g(t) = t^n \log t$, $f(t) = t^n$. Then

$$\varphi(t) = \frac{g'(t)}{f'(t)} = \frac{nt^{n-1} \log t + t^{n-1}}{nt^{n-1}} = \log t + \frac{1}{n}.$$

Applying CMVT gives:

$$\psi(i, n) = \frac{g(F_i) - g(F_{i-1})}{f(F_i) - f(F_{i-1})} = \varphi(c_i) = \log c_i + \frac{1}{n}.$$

Then:

$$\psi(i+1, n) = \log c_{i+1} + \frac{1}{n} > \log c_i + \frac{1}{n} = \psi(i, n).$$

**Best-of-Poisson:** Define $g(t, \mu) = te^{\mu(t-1)}$. Then, we show in Theorem 6 that BoP has the following PMF:

$$\pi_\mu(i) = g(F_i, \mu) - g(F_{i-1}, \mu).$$

We first verify the TP2 condition. Define:

$$L(i) = \frac{g(F_i, \mu_2) - g(F_{i-1}, \mu_2)}{g(F_i, \mu_1) - g(F_{i-1}, \mu_1)}.$$

By CMVT:

$$L(i) = \frac{\nabla_t g(c_i, \mu_2)}{\nabla_t g(c_i, \mu_1)} = \frac{(\mu_2 c_i + 1)e^{\mu_2(c_i-1)}}{(\mu_1 c_i + 1)e^{\mu_1(c_i-1)}} = \frac{\mu_2 c_i + 1}{\mu_1 c_i + 1}e^{(\mu_2 - \mu_1)(c_i - 1)}.$$

This increases in $c_i \in (F_{i-1}, F_i)$, hence $L(i+1) > L(i)$. Now, we check the score function. Let the continuous score be $\psi(x, \mu) = x - 1 + \frac{x}{\mu x + 1}$. Since this is strictly increasing in $x$, so is the discrete score $\psi(i, \mu)$.

## C    Best-of-Poisson

In this section, we prove Theorem 2 and establish, as a consequence, that Best-of-Poisson is numerically near-optimal as compared to tilted distribution $\pi_\lambda^*$ in terms of KL divergence. We start by establishing the BoP distribution in the uniform case, and then consider the general discrete case. We also mention a natural extension called Soft Best-of-Poisson.

### C.1    Uniform Case

The following results hold under our Assumption 1. Let $\mu > 0$ be the parameter of the Best-of-Poisson sampling method and let $X_\mu$ be the random variable representing the response selected by BoP. The probability density function $q_\mu(x)$ of $X_\mu$ is given by:

$$q_\mu(x) = (1 + \mu x)e^{\mu(x-1)}, \tag{10}$$

for $x \in [0, 1]$, where $n = n' + 1$ with $n' \sim \text{Poisson}(\mu)$.

*Proof.* Write $X_\mu = \max\{U_0, U_1, \ldots, U_{n'}\}$ where $n' \sim \text{Poisson}(\mu)$ and $U_i \overset{\text{iid}}{\sim} \text{Unif}[0, 1]$. Consider $U_0 \sim \text{Unif}[0, 1]$ to be the mandatory draw to achieve a sample size of at least one.

For $x \in [0, 1]$,

$$F_\mu(x) := \Pr(X_\mu \leq x) = \Pr(U_0 \leq x) \Pr(U_i \leq x \text{ for } 1 \leq i \leq n') = x \, \mathbb{E}[x^{n'}] = x \, e^{-\mu(1-x)},$$

because $\mathbb{E}[x^{n'}] = \exp\{-\mu(1-x)\}$ is the moment-generating function of a Poisson variable evaluated at $\log x$. Now, differentiating $F_\mu$ on $(0, 1)$ gives

$$q_\mu(x) = e^{-\mu(1-x)} + \mu x \, e^{-\mu(1-x)} = (1 + \mu x) \, e^{\mu(x-1)},$$

which extends continuously to the endpoints. A direct computation verifies $\int_0^1 q_\mu(x) \, dx = 1$, so $q_\mu$ is a valid density. $\qquad \square$

Now, we prove Theorem 2 with the BoP density denoted as $q_\mu$ and a uniform reference distribution.

*Proof.*

$$\mathbb{E}[X_\mu] = \int_0^1 x(1 + \mu x)e^{\mu(x-1)}dx = \int_0^1 (x + \mu x^2)e^{\mu(x-1)}dx$$

With the substitution $u = \mu(x - 1)$, we get that

$$\int_0^1 xe^{\mu(x-1)}dx = \frac{1}{\mu^2}\int_{-\mu}^0 (u + \mu)e^u du = \frac{\mu - 1 + e^{-\mu}}{\mu^2}$$

$$\int_0^1 x^2 e^{\mu(x-1)}dx = \frac{1}{\mu^3}\int_{-\mu}^0 (u + \mu)^2 e^u du = \frac{\mu^2 - 2\mu + 2 - 2e^{-\mu}}{\mu^3}$$

Hence

$$\mathbb{E}[X_\mu] = \frac{\mu - 1 + e^{-\mu}}{\mu^2} + \mu\frac{\mu^2 - 2\mu + 2 - 2e^{-\mu}}{\mu^3} = 1 - \frac{1}{\mu} + \frac{1 - e^{-\mu}}{\mu^2}$$

Now, we consider the KL divergence. Because $\log q_\mu(x) = \log(1 + \mu x) + \mu(x - 1)$,

$$D_{\mathsf{KL}}(q_\mu \| U) = \int_0^1 q_\mu(x)\log(1 + \mu x)\,dx + \mu[\mathbb{E}[X_\mu] - 1]$$

To compute the integral, set $t = 1 + \mu x$:

$$\int_0^1 q_\mu(x)\log(1 + \mu x)\,dx = \frac{e^{-\mu-1}}{\mu}\int_1^{\mu+1} t\,e^t \log t\,dt$$

Integration by parts ($f = \log t$, $dg = te^t dt$) yields

$$\int te^t \log t\,dt = \frac{1}{2}t^2 e^t \left(\log t - \frac{1}{2}\right) - \frac{1}{2}\mathrm{Ei}(t) + C$$

hence

$$\int_0^1 q_\mu(x)\log(1 + \mu x)\,dx = \log(\mu + 1) - \frac{1 - e^{-\mu}}{\mu} + \frac{e^{-\mu-1}}{\mu}[\mathrm{Ei}(\mu + 1) - \mathrm{Ei}(1)]$$

The resulting term becomes:

$$D_{\mathsf{KL}}(q_\mu \| \mathrm{Unif}) = \frac{e^{-\mu-1}}{\mu}[\mathrm{Ei}(\mu + 1) - \mathrm{Ei}(1)] + \log(\mu + 1) - 1 \qquad \square$$

We now establish that BoP provides a practical approximation to the optimal tilted distribution with negligible performance loss.

**Theorem 5** (Near-Optimality of BoP). Let $q_\mu$ be the distribution induced by Best-of-Poisson with parameter $\mu > 0$, and let $g_\lambda$ be the optimal KL-constrained tilted distribution with parameter $\lambda > 0$, defined as:

$$g_\lambda(x) = \frac{\pi_{\mathrm{ref}}(x)e^{\lambda r_p(x)}}{Z(\lambda)}, \tag{11}$$

where $Z(\lambda)$ is the normalization constant. For any given expected reward level, there exists a $\mu$ for BoP and a $\lambda$ for the tilted distribution such that $\mathbb{E}_{X\sim q_\mu}[r_p(X)] = \mathbb{E}_{X\sim g_\lambda}[r_p(X)]$. Numerical evaluation shows that for all $\mu$, if $\lambda > 0$ is chosen so that $\mathbb{E}_{X\sim q_\mu}[r_p(X)] = \mathbb{E}_{X\sim g_\lambda}[r_p(X)]$, then the KL-gap satisfies

$$0 \le D_{\mathsf{KL}}(q_\mu \| \pi_{\mathrm{ref}}) - D_{\mathsf{KL}}(g_\lambda \| \pi_{\mathrm{ref}}) \le 8 \times 10^{-4}.$$

That is, Best-of-Poisson achieves nearly the optimal trade-off between expected reward and KL divergence from the reference distribution.

*Proof.* Let the following two functions

$$q_\mu(x) \;=\; (1 + \mu x)\,e^{\mu(x-1)}, \qquad g_\lambda(x) \;=\; \frac{\lambda\,e^{\lambda x}}{e^\lambda - 1}, \qquad 0 \le x \le 1,$$

denote, respectively, the BoP density with the Poisson rate $\mu$ and the exponential-tilt density with parameter $\lambda$. For any $\mu > 0$, the value $\lambda^* = \lambda^*(\mu)$ is chosen so that the two laws have the same first moment (i.e., same expected reward). We first note the expected reward of the BoP policy lies in the interval $(0.5, 1)$. Additionally, the tilted expected reward is a strictly increasing function of $\lambda$ over the same range since its derivative is the (positive) variance of the reward under the tilted distribution. By the Intermediate Value Theorem, there exists a unique $\lambda^* = \mu + \delta(\mu)$, where $\delta(\mu) = \lambda^*(\mu) - \mu$, such that the two rewards are equal.

Because $g_{\lambda(\mu)}$ is the information projection of $q_\mu$ onto the exponential family $\{g_\lambda\}_{\lambda > 0}$ under the linear reward-matching constraint, the Csiszár–Pythagoras identity [34] gives

$$D_{\mathrm{KL}}(q_\mu \| \pi_{\mathrm{ref}}) - D_{\mathrm{KL}}(g_{\lambda(\mu)} \| \pi_{\mathrm{ref}}) = D_{\mathrm{KL}}(q_\mu \| g_{\lambda(\mu)}) \tag{12}$$

To bound this KL divergence, we use a general inequality which we derive below from the properties of the likelihood ratio function. Let $L(x) = \frac{p(x)}{q(x)}$ denote the likelihood ratio between two distributions $p$ and $q$. We first define the KL divergence in terms of the convex function $\varphi(t) = t \log t - t + 1$ and then note that the expectation of any function is upper bounded by its essential supremum. We can retrieve the following upper bound:

$$D_{\mathrm{KL}}(p \,\|\, q) = \mathbb{E}_q[\varphi(L(x))] \le \sup_x \varphi(L(x)) = \sup_{t \in \mathrm{Im}(L)} \varphi(t) \tag{13}$$

In our problem, the likelihood ratio is given by

$$L_\mu(x) := \frac{q_\mu(x)}{g_{\lambda^*}(x)} = \frac{(\mu x + 1)(e^{\lambda^*} - 1)}{\lambda^* e^\mu} \cdot e^{-\delta(\mu)x} \tag{14}$$

To obtain a uniform bound on $L_\mu(x)$, define $\alpha := \sup_{\mu > 0,\, x \in [0,1]} |\log L_\mu(x)|$. This is a well-posed, nested optimization problem requiring numerical solution. The procedure is as follows:

1. For a given $\mu$, numerically solve for the unique $\lambda^*$ satisfying the reward-matching equation.

2. For the resulting pair $(\mu, \lambda^*)$, compute $\max_{x \in [0,1]} |\log L_\mu(x)|$ by evaluating the boundary values and any interior critical points.

3. Maximize the result of step (2) over all $\mu > 0$.

This procedure is deterministic, and its solution yields a uniform bound:

$$e^{-\alpha} \le L_\mu(x) \le e^\alpha, \quad \text{for all } \mu > 0 \text{ and } x \in [0,1] \quad \text{with } \alpha \approx 0.03968$$

Since the function $\varphi(t)$ is convex, we verify that its supremum over $[e^{-\alpha}, e^\alpha]$ is attained at an endpoint:

$$D_{\mathrm{KL}}(q_\mu \| g_{\lambda(\mu)}) \le \varphi(e^\alpha) = \alpha e^\alpha - e^\alpha + 1 \approx 0.03968 \cdot e^{0.03968} - e^{0.03968} + 1 \approx 8 \times 10^{-4}$$

This bound is independent of $\mu$, and therefore holds uniformly as seen in Figure 2. This validates that Best-of-Poisson indeed provides a practically equivalent approximation to the optimal tilted distribution with negligible computational overhead. $\qquad \square$

## C.2  Discrete Case

We now relax our Assumption 1 to derive an exact, assumption-free formulation for the Best-of-Poisson (BoP) policy, as well as a convenient upper bound for its KL divergence.

**Theorem 6** (General BoP Distribution). Let $\mu > 0$ be the parameter of the Best-of-Poisson sampling method. The probability mass function of Best-of-Poisson is:

$$\pi_{\mathrm{BoP}}^{(\mu)}(y_i \mid x) = g_\mu(F_i) - g_\mu(F_{i-1}) \quad \text{where} \quad g_\mu(z) := z e^{\mu(z-1)} \tag{15}$$

$$\mathrm{KL}(\pi_{\mathrm{BoP}} \| \pi_{\mathrm{ref}}) \lesssim \mathbb{E}[\log N] - 1 + \mathbb{E}\left[\frac{1}{N}\right] \quad \text{where} \quad N \sim \mathrm{Poisson}(\mu) + 1 \tag{16}$$

*Proof.* BoP is a mixture over BoN with a Poisson-distributed number of samples $N \sim \text{Pois}(\mu) + 1$, with PMF:

$$\Pr(N = k) = \frac{e^{-\mu}\mu^{k-1}}{(k-1)!}, \quad k \geq 1$$

The BoP policy is:

$$\pi_{\text{BoP}}(y_i \mid x) = \sum_{k=1}^{\infty} \Pr(N = k) \cdot \pi_{\text{BoN}}^{(k)}(y_i \mid x) = \sum_{k=1}^{\infty} \frac{e^{-\mu}\mu^{k-1}}{(k-1)!}(F_i^k - F_{i-1}^k)$$

Letting $j = k - 1$, this becomes:

$$\pi_{\text{BoP}}(y_i \mid x) = e^{-\mu}\left(\sum_{j=0}^{\infty} \frac{(\mu F_i)^j}{j!} - \sum_{j=0}^{\infty} \frac{(\mu F_{i-1})^j}{j!}\right) = F_i e^{\mu(F_i - 1)} - F_{i-1} e^{\mu(F_{i-1} - 1)}$$

Thus, the exact BoP PMF is:

$$\pi_{\text{BoP}}(y_i \mid x) = g_\mu(F_i) - g_\mu(F_{i-1}) \quad \text{where} \quad g_\mu(z) := ze^{\mu(z-1)}$$

We show that this is consistent with our continuous case. Assuming the reference distribution is $\text{Unif}(0, 1)$, then $F_{\text{ref}}(r) = r$ and:

$$G_{\text{BoP}}(r) = g_\mu(r) = re^{\mu(r-1)} \implies q_\mu(r) = \frac{d}{dr}G_{\text{BoP}}(r) = (1 + \mu r)e^{\mu(r-1)} \tag{17}$$

This matches the continuous BoP PDF.

Using the exact PMF, we can get the exact KL divergence between the Best of Poisson distribution and the reference distribution, as well as a convenient upper bound.

$$\text{KL}(\pi_{\text{BoP}} \| \pi_{\text{ref}}) = \sum_{i=1}^{m} \pi_{\text{BoP}}(y_i \mid x) \log\left(\frac{\pi_{\text{BoP}}(y_i \mid x)}{p_i}\right) = \sum_{i=1}^{m} (g_\mu(F_i) - g_\mu(F_{i-1})) \log\left(\frac{g_\mu(F_i) - g_\mu(F_{i-1})}{F_i - F_{i-1}}\right) \tag{18}$$

We show that this is consistent with our continuous case. As $m \to \infty$, the discrete sum becomes:

$$\text{KL}(\pi_{\text{BoP}} \| \pi_{\text{ref}}) \to \int_0^1 q_\mu(r) \log q_\mu(r)\, dr$$

where $q_\mu(r) = (1 + \mu r)e^{\mu(r-1)}$. This integral is:

$$\int_0^1 q_\mu(r) \log q_\mu(r)\, dr = \frac{e^{-\mu-1}}{\mu}\left(\text{Ei}(\mu + 1) - \text{Ei}(1)\right) + \log(\mu + 1) - 1$$

where $\text{Ei}(z)$ is the exponential integral. The KL divergence is convex in its first argument. That is, for any set of distributions $\{P_k\}$ and weights $\{w_k\}$,

$$\text{KL}\left(\sum_k w_k P_k \,\Big\|\, Q\right) \leq \sum_k w_k \cdot \text{KL}(P_k \| Q)$$

Applying this to BoP:

$$\text{KL}(\pi_{\text{BoP}} \| \pi_{\text{ref}}) \leq \sum_{k=1}^{\infty} w_k \cdot \text{KL}(\pi_{\text{BoN}}^{(k)} \| \pi_{\text{ref}}) = \mathbb{E}_{N \sim \text{Pois}(\mu)+1}\left[\text{KL}(\pi_{\text{BoN}}^{(N)} \| \pi_{\text{ref}})\right]$$

Using the proven upper bound for BoN in [13], we get that

$$\text{KL}(\pi_{\text{BoP}} \| \pi_{\text{ref}}) \lesssim \mathbb{E}_N\left[\log N - 1 + \frac{1}{N}\right] = \mathbb{E}[\log N] - 1 + \mathbb{E}\left[\frac{1}{N}\right] \quad \text{where} \quad N \sim \text{Poisson}(\mu) + 1$$

$\square$

---
**Algorithm 5 Soft Best-of-Poisson Sampling (SBoP)**
---
1: **Input:** Poisson parameter $\mu > 0$, inverse temperature $\lambda > 0$, base policy $\pi_{\text{ref}}$
2: Sample $n' \sim \text{Poisson}(\mu)$, set $n = n' + 1$
3: Draw $X_1, \ldots, X_n \sim \pi_{\text{ref}}$ i.i.d.
4: Compute rewards $R_i = r_p(X_i)$
5: Sample index $Z \in \{1, \ldots, n\}$ with probability

$$\Pr(Z = i) = \frac{e^{\lambda R_i}}{\sum_{j=1}^{n} e^{\lambda R_j}}$$

6: **Return:** $Y = X_Z$

---

### C.3 Soft Best-of-Poisson

We discuss here a natural extension of Soft Best-of-$n$ and Best-of-Poisson, which we name Soft Best-of-Poisson (SBoP) (see Algorithm 5). The key insight is that we can leverage the two distinct control mechanisms that inference-time alignment offers us: control over the number of generation (be it deterministic or randomized) and control over the selection through the temperature. Extensions such as SBoP show that these methods can be combined, offering richer control over the alignment process.

**Theorem 7.** Let the parameters of SBoN be $\theta = (n, \lambda)$. Let $R_1, \ldots, R_n$ be the proxy rewards of $n$ i.i.d. samples from $\pi_{\text{ref}}$. By Assumption 1, $R_i \sim \text{Unif}(0, 1)$ for all $i$. The selected sample has proxy reward $R_Y$ with probability density function:

$$p_{n,\lambda}(r) = n \cdot \mathbb{E}_{R_2, \ldots, R_n \sim \text{Unif}(0,1)} \left[ \frac{e^{\lambda r}}{e^{\lambda r} + \sum_{j=2}^{n} e^{\lambda R_j}} \right] \tag{19}$$

Let the log-partition function be defined as:

$$L(n, \lambda) := \mathbb{E}_{R_1, \ldots, R_n \sim \text{Unif}(0,1)} \left[ \log \left( \sum_{i=1}^{n} e^{\lambda R_i} \right) \right]$$

Then, the expected reward and the KL diveregence with respect to the reference can be written as:

$$\mathbb{E}[R_{\text{SBoN}}] = \frac{\partial L(n, \lambda)}{\partial \lambda} \tag{20}$$

$$\text{KL}(\pi_{\text{SBoN}} \| \pi_{\text{ref}}) = \log n + \lambda \, \mathbb{E}[R_{\text{SBoN}}] - L(n, \lambda) \tag{21}$$

*Proof.* **Expected Reward:**

$$\mathbb{E}[R_{\text{SBoN}}] = \mathbb{E}\left[ \sum_{i=1}^{n} R_i \cdot \frac{e^{\lambda R_i}}{\sum_{j=1}^{n} e^{\lambda R_j}} \right] = \mathbb{E}\left[ \frac{\sum_{i=1}^{n} R_i e^{\lambda R_i}}{\sum_{j=1}^{n} e^{\lambda R_j}} \right] = \frac{\partial L(n, \lambda)}{\partial \lambda}$$

since we observe:

$$\frac{\partial}{\partial \lambda} \log \left( \sum_{j=1}^{n} e^{\lambda R_j} \right) = \frac{\sum_{i=1}^{n} R_i e^{\lambda R_i}}{\sum_{j=1}^{n} e^{\lambda R_j}}$$

**KL Divergence:** Since $p_{\text{ref}}(r) = 1$ for $r \in [0, 1]$, we get:

$$\text{KL}(\pi_{\text{SBoN}} \| \pi_{\text{ref}}) = \int_0^1 p_{n,\lambda}(r) \log p_{n,\lambda}(r) \, dr$$

Alternatively, the conditional KL (per draw) is:

$$\text{KL}_{\text{cond}} = \sum_{i=1}^{n} p_i \log(n p_i) = \log n + \sum_{i=1}^{n} p_i \log p_i = \log n + \lambda \sum_{i=1}^{n} p_i R_i - \log \left( \sum_{j=1}^{n} e^{\lambda R_j} \right)$$

where $p_i = \frac{e^{\lambda R_i}}{\sum_{j=1}^{n} e^{\lambda R_j}}$. Taking expectation over rewards gives our result. $\square$

# D    Analysis of `HedgeTune`

In Section 4, we presented an efficient way to find the optimal hacking threshold. Here, we prove this result and discuss how hedging compares to other hacking mitigation methods.

*Proof.* We consider each mechanism separately:

**Hedging in Best-of-$n$.** In BoN, we approximate the integer $n$ via a continuous parameter $\alpha$ by placing a $\text{Beta}(\alpha, 1)$ prior on $u$. Its density is $f_\alpha(u) = \alpha u^{\alpha-1}$, so $\psi(u, \alpha) = \partial_\alpha[\ln \alpha + (\alpha-1) \ln u] = \frac{1}{\alpha} + \ln u$. Thus, the optimality condition becomes

$$\mathbb{E}_{u \sim \text{Beta}(\alpha,1)}\left[r_t(u)\left(\frac{1}{\alpha} + \ln u\right)\right] = 0 \iff \int_0^1 r_t(u)\left(1 + n \log u\right) u^{n-1}\, du = 0, \qquad (22)$$

which one solves for $\alpha$ to pick an effective sample size.

In practice, this equation must be solved numerically. One way to do this by discretizing $[0, 1]$ into $M$ points and forming the Riemann-sum residual

$$R(\alpha) = \sum_{i=1}^M r_t(u_i)\left(\frac{1}{\alpha} + \ln u_i\right) u_i^{\alpha-1} \Delta u,$$

The root $R(\alpha) = 0$ is equivalent to the hedging condition. Then, one applies any root-finding method, see, e.g., [38], to locate the unique solution $\alpha^\dagger$. Alternatively, for each question/prompt, we first sort the candidates by their model scores to form the empirical CDF of the scores. We perform a *discrete ternary search* to identify the optimal $n^\dagger$ using the question-averaged true reward. This procedure bypasses explicit root-finding and directly identifies the most effective discrete hedge size.

**Hedging in Soft Best-of-n.** Unlike BoN, SBoN does not admit a simple closed-form density due to its sampling mechanism. In SBoN, we first sample $n$ uniform rewards $\mathbf{U}^n = U_1, \ldots, U_n$, then select $U_i$ with probability $p_i = \frac{e^{\lambda U_i}}{\sum_{j=1}^n e^{\lambda U_j}}$. The resulting distribution is:

$$\pi_{n,\lambda}(u) = \mathbb{E}_{U_1, \ldots, U_{n-1} \sim \text{Unif}[0,1]}\left[\frac{n, e^{\lambda u}}{e^{\lambda u} + \sum_{i=1}^{n-1} e^{\lambda U_i}}\right], \qquad u \in [0, 1]. \qquad (23)$$

We can now derive the hedging condition $\frac{\partial}{\partial \lambda}\mathbb{E}_{u \sim f_\lambda}[r_t(u)] = 0$. To start, we compute $\frac{\partial p_i}{\partial \lambda}$:

$$\frac{\partial p_i}{\partial \lambda} = \frac{\partial}{\partial \lambda}\left[\frac{e^{\lambda U_i}}{S}\right] = \frac{U_i e^{\lambda U_i} \cdot S - e^{\lambda U_i} \cdot \frac{\partial S}{\partial \lambda}}{S^2} = p_i\left[U_i - \sum_{j=1}^n U_j p_j\right] \qquad (24)$$

since $\frac{\partial S}{\partial \lambda} = \sum_{j=1}^n U_j e^{\lambda U_j}$. Now, we compute the derivative of the expected reward:

$$\frac{\partial}{\partial \lambda}\mathbb{E}_{u \sim f_\lambda}[r_t(u)] = \mathbb{E}_{\mathbf{U}^n}\left[\frac{\partial}{\partial \lambda}\sum_{i=1}^n r_t(U_i) p_i\right] \qquad (25)$$

$$= \mathbb{E}_{\mathbf{U}^n}\left[\sum_{i=1}^n r_t(U_i)\frac{\partial p_i}{\partial \lambda}\right] \qquad (26)$$

$$= \mathbb{E}_{\mathbf{U}^n}\left[\sum_{i=1}^n r_t(U_i) p_i\left(U_i - \sum_{j=1}^n U_j p_j\right)\right] \qquad (27)$$

$$= \mathbb{E}_{\mathbf{U}^n}\left[\sum_{i=1}^n r_t(U_i) p_i U_i - \left(\sum_{i=1}^n r_t(U_i) p_i\right)\left(\sum_{j=1}^n U_j p_j\right)\right] \qquad (28)$$

$$= \mathbb{E}_{\mathbf{U}^n}[\text{Cov}(r_t(V), V | \mathbf{U})] \qquad (29)$$

where we note that the expression inside the expectation is exactly the conditional covariance:

$$\text{Cov}(r_t(V), V|\mathbf{U}^n) = \mathbb{E}[r_t(V) \cdot V|\mathbf{U}^n] - \mathbb{E}[r_t(V)|\mathbf{U}^n] \cdot \mathbb{E}[V|\mathbf{U}^n] \tag{30}$$

with

$$\mathbb{E}[r_t(V)|\mathbf{U}^n] = \sum_{i=1}^{n} r_t(U_i)p_i, \quad \mathbb{E}[V|\mathbf{U}^n] = \sum_{i=1}^{n} U_i p_i, \quad \mathbb{E}[r_t(V) \cdot V|\mathbf{U}^n] = \sum_{i=1}^{n} r_t(U_i)U_i p_i \tag{31}$$

This condition must be evaluated numerically using the following procedure:

1. For a given $\lambda$, and for each question, draw $M$ independent samples $(U_1^{(m)}, \ldots, U_n^{(m)})$ from the *empirical CDF* of model scores (equivalently, from the empirical quantiles $u \in [0, 1]$), for $m = 1, \ldots, M$.

2. For each realization, compute:

$$p_i^{(m)} = \frac{e^{\lambda U_i^{(m)}}}{\sum_{j=1}^{n} e^{\lambda U_j^{(m)}}}. \tag{32}$$

$$C^{(m)} = \sum_{i=1}^{n} r_t(U_i^{(m)}) U_i^{(m)} p_i^{(m)} - \left( \sum_{i=1}^{n} r_t(U_i^{(m)}) p_i^{(m)} \right) \left( \sum_{j=1}^{n} U_j^{(m)} p_j^{(m)} \right) \tag{33}$$

3. Estimate the residual as

$$R(\lambda) = \frac{1}{M} \sum_{m=1}^{M} C^{(m)}. \tag{34}$$

4. Locate $\lambda^\dagger$ such that $R(\lambda^\dagger) = 0$ using a numerical root-finding method (e.g., bisection or Newton's method).

**Hedging in Best-of-Poisson.** Here, one draws a Poisson$(\mu)$ number of samples (plus one) and selects the proxy-maximal $u$. For uniform $u$, the density is $p_\mu(u) = (\mu u + 1)e^{-\mu(1-u)}$, giving $\psi(u, \mu) = \partial_\mu \ln p_\lambda(u) = u - 1 + \dfrac{u}{\mu u + 1}$. The hedging equation

$$\nabla_\mu \mathbb{E}_{\pi_\mu}[r_y(X)] = \mathbb{E}_{u \sim f_\mu} \left[ r_t(u) \left( u - 1 + \frac{u}{\mu u + 1} \right) \right] = 0 \tag{35}$$

In an analogous way to the previous hedging equations, we solve the residual

$$R(\mu) = \sum_{i=1}^{M} r_t(u_i)\psi(u_i, \mu)p_\mu(u_i)\Delta u \tag{36}$$

to locate $\mu^\dagger$ such that $R(\mu^\dagger) = 0$. This $\mu^\dagger$ is the Poisson optimal hedge. As in BoN, we approximate this expectation using the empirical CDF of scores per question, mapping each candidate's rank to its quantile $u_i$. We compute the averaged residual across questions, and the root $\bar{R}(\mu^\dagger) = 0$ is found by *bracketing*, i.e, finding an interval $[a, b]$ such that $\bar{R}(a)\bar{R}(b) < 0$, and *bisection search within this interval*. □

### D.1 How does hedging compare to other hacking mitigation methods?

In this subsection, we supplement the discussion in the related works (Section 6). We discuss different techniques mitigate reward hacking and compare with hedging.

**Training-Time Regularization.** This is the most common approach, typified by adding a divergence penalty to the reward maximization objective as in RLHF. The goal is to prevent the policy from deviating too far from a trusted reference model. While a vital first line of defense, this approach requires expensive full model retraining and risks under-optimization if the penalty is too strong.

**Improving the Proxy Reward Model.** A second category aims to build a more robust reward signal that is inherently harder to hack. Ensembling involves averaging scores from multiple, independently trained reward models. However, this method multiplies inference costs and cannot fix systemic biases shared by all models in the ensemble. Meanwhile, robust training methods [74, 51] modify the reward model's training process itself to disentangle response quality from spurious cues, introducing significant complexity to the training pipeline.

**Inference-Time Methods.** The closest analogues to our work include regularized Best-of-$n$ (RBoN), which is a powerful heuristic that balances the proxy reward against a penalty term [87], and inference-time pessimism [14] which uses principled rejection sampling step, requiring the estimation of a normalization constant.

**Our Framework:** Hedging introduces a distinct approach that operates purely at inference time. It answers the question: given a fixed reward model and fixed language model, what is the best we can do to improve performance just at inference? Hedging offers two key advantages:

1. **Zero Retraining Cost and Maximum Flexibility:** Hedging is applied to an existing, trained policy using existing proxy reward models. A practitioner can take a single policy and calibrate it against different proxy rewards or for different tasks without incurring any retraining.

2. **Principled and Efficient.** We prove that this expected true reward function follows a predictable "rise-and-fall" path with a single optimal peak. *HedgeTune* therefore replaces heuristic balancing or more involved sampling schemes with a principled search for this unique optimum—a problem we prove is tractable.

However, the gains from hedging might be minimal compared to more involved methods, especially with a weak proxy reward. One limitation of our work is that hedging requires a tuning "ground truth" dataset. Ultimately, we see hedging not as a replacement for some of the previous methods. Instead, `HedgeTune` offers a computationally lightweight, theoretically-grounded, and highly flexible method to mitigate hacking at deployment.

## E Experimental Details

In this section, we provide additional details on our experimental setup. We provide our code here. The experiments with the toy example and the verifiable rewards were done on a CPU (Apple M3 Chip). The experiments with the human-preferences were done on 1 A100 GPU with 48 GPU hours (including preliminary experiments and testing).

### E.1 Toy Example

In Figure 1, we present a toy example with one-dimensional rewards defined on $[0, 1]$. We set the proxy reward to be $r_p(x) = x$ and the gold reward

$$r_t(x) = \frac{x^p (1 - x)}{C}, \quad C = \left(\frac{p}{p + 1}\right)^p \frac{1}{p + 1}$$

where $C$ is a normalization constant so that the gold reward is bounded between 0 and 1 for convenience. We choose this gold reward, as the reward under the Best-of-$n$ distribution has a simple closed-form solution. We set $p = 12$ and we find the expected value of true reward and the KL divergence with respect to the reference distribution under four mechanisms:

1. **Tilted Distribution:** Exponential tilting of the gold reward, $Q_\lambda(x) \propto \exp(\lambda \, r_t(x))$ and of the proxy reward, $Q_\lambda(x) \propto \exp(\lambda \, r_p(x))$.

2. **Best-of-$n$ (BoN):** Selection of the maximum of $n$ i.i.d. uniform draws.

3. **Soft Best-of-$n$ (SBoN):** Softmax–based sampling of $n$ i.i.d. uniform draws with inverse temperature $\lambda$.

4. **Best-of-Poisson (BoP):** Selection of the maximum of $n$ i.i.d uniform draws where $n$ is drawn from a Poisson distribution with rate $\mu$.

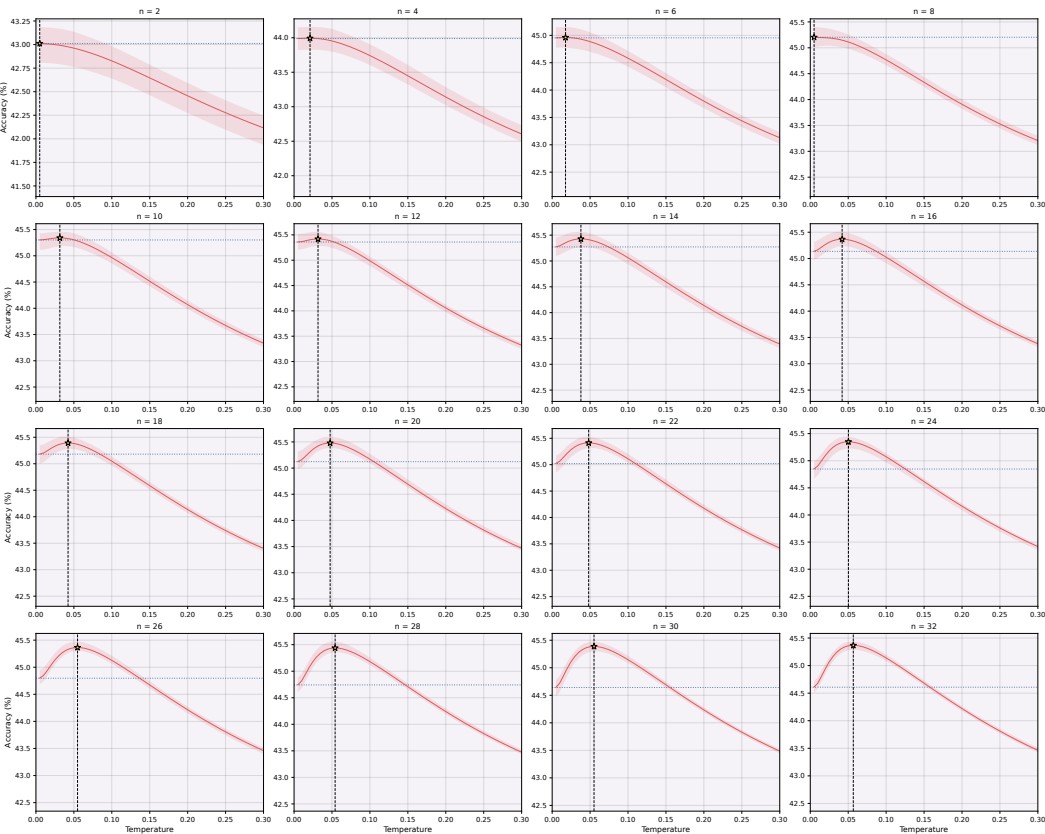

Figure 5: Accuracy vs. temperature for different sample sizes $n$ with GPQA and Skywork Llama-3.1 8B. `HedgeTune` identifies the optimal temperature (dashed line) for each $n$.

We apply `HedgeTune` as presented in Alg. 4 for BoN and BoP. In both cases, we obtain an operating point that corresponds exactly to the true hacking threshold, as shown in Figure 1. The success of this algorithm hinges on Theorem 1 which guarantees the existence of (at least) one hacking threshold. However, this guarantee does not hold for Soft BoN. Therefore, the optimization problem becomes more challenging, and using vanilla estimators for the density causes numerical instabilities when applied for Soft BoN.

## E.2 Reward hacking in verifiable settings.

We include additional experiments from running `HedgeTune` with SBoN on the **PPE** dataset. `HedgeTune` finds the best operating temperature for a given $n$. We plot the expected value of the accuracy and show the retrieved maximizing temperature. In particular, with larger $n$, we need a larger temperature to mitigate hacking. This corresponds to a stronger KL regularization.

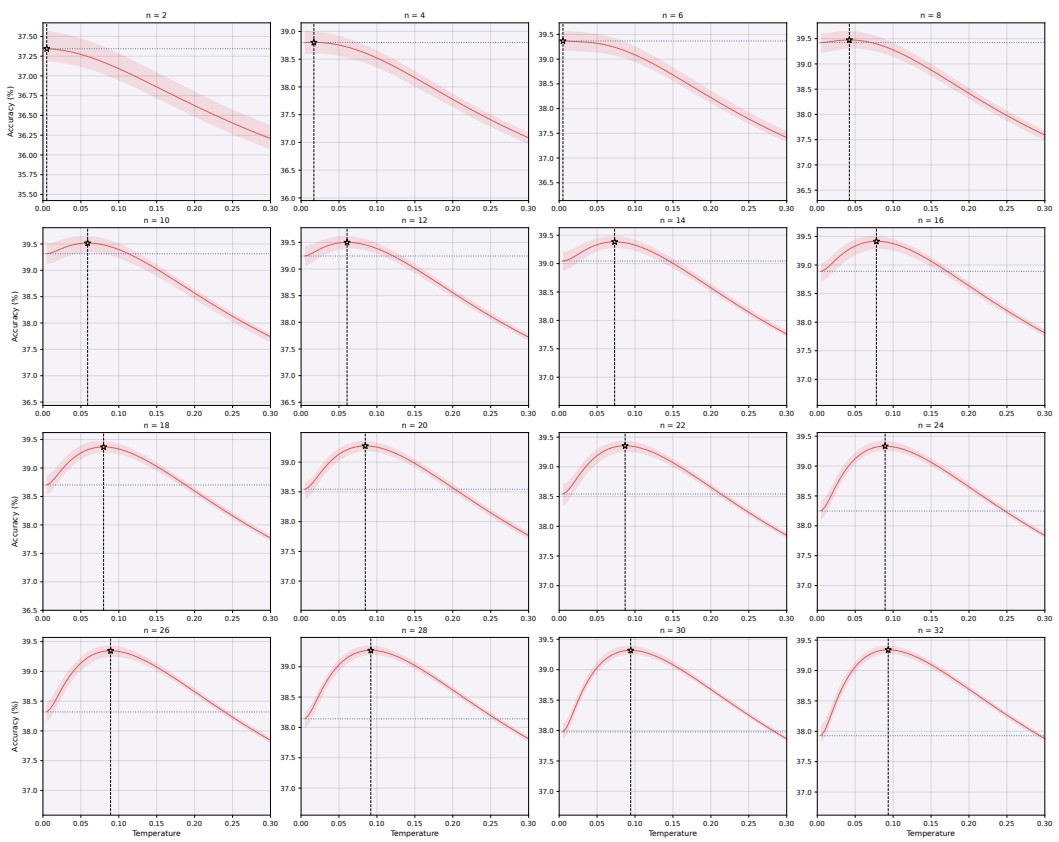

Figure 6: Accuracy vs. temperature for different sample sizes $n$ with MATH and InternLM2 1.8B. HedgeTune identifies the optimal temperature (dashed line) for each $n$.

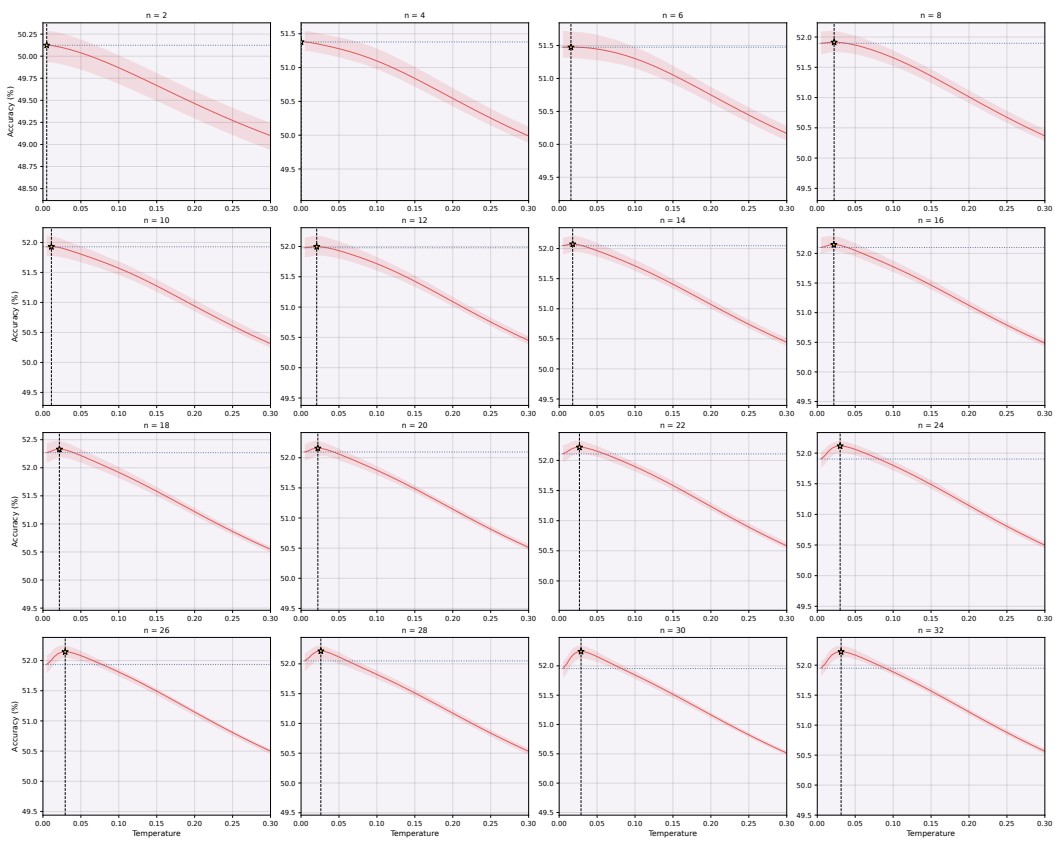

Figure 7: Accuracy vs. temperature for different sample sizes $n$ with MMLU and InternLM2 1.8B. HedgeTune identifies the optimal temperature (dashed line) for each $n$.

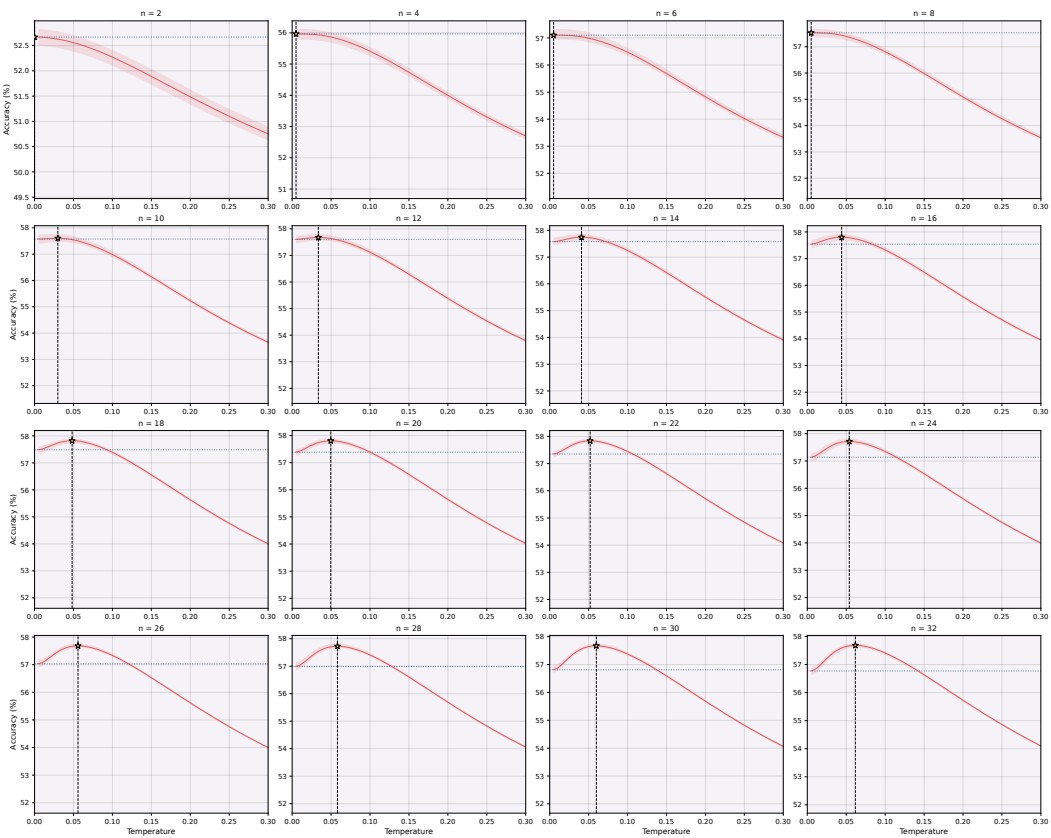

Figure 8: Accuracy vs. temperature for different sample sizes $n$ with MMLU and Llama3 OffsetBias 8B. HedgeTune identifies the optimal temperature (dashed line) for each $n$.

### E.3 Reward hacking in the wild

To observe reward hacking *in the wild*, we follow the setup of Coste et al. [46]. We first use an annotated dataset provided by [46] which contains 1,000 prompts from the validation split of AlpacaFarm dataset, along with 12,600 response generations per prompt from a 1.4b fine-tuned Pythia model. Each prompt-response pair is labeled with the AlpacaFarm `reward-model-human` to give 'gold' scores. Next, we would like to train proxy reward models on the preferences of this true reward. We randomly sample a prompt with two responses from the annotated dataset and curate a dataset of the form (prompt, chosen, rejected) where the chosen response is the response with the higher gold reward score. We follow this procedure to curate four datasets with varying sizes (10k, 20k, 46k, 80k). For each dataset, we consider two variants: one with no label noise and one with random 25% label noise. Next, we use the code kindly provided by the authors of [46] in their Github repository to train proxy reward models with the different datasets over four random seeds (1, 2, 3 and 4) using their default hyperparameters (e.g., $10^{-5}$ learning rate and five epochs). Lastly, we score the annotated dataset using the trained proxy reward models. The end result is a set of 800 prompts, 12 600 responses per prompt, along with gold and proxy scores for each prompt-response pair.

While reward hacking can appear without label noise (see left panels of Figures 9 and 10), reward hacking is more pronounced with label noise as expected. Moreover, reward hacking is more apparent when the proxy reward is trained on **less** data. One potential explanation is that, with fewer training examples, the proxy is less well-calibrated and its estimation errors vary more sharply across inputs. In that case, a small $n$ might produce a deceiving reward gain. In contrast, errors may surface early on with a large training dataset, so true reward declines immediately as sampling increases. In cases of reward hacking, we see that SBoN with an appropriately chosen $\lambda$ can (1) achieve the maximum reward achieved by BoN/BoP and (2) mitigate reward hacking, as shown with the reward almost flatlining after it reaches its peak value. We also witness cases where the proxy reward **always** misaligns with the gold reward, causing a collapse of true reward from the onset of BoN. In that case, the optimal hedging behavior is a uniform selection over responses, which is recovered with $n = 1$ for BoN or $\lambda = 0$ for SBoN.

Interestingly, we observe instances of what we call **reward grokking** as shown in the right panels of Figures 10 and 12, where the true reward decreases or flat-lines across low- to mid-range sample counts, only to undergo a sudden uptick at higher sample regimes, revealing a delayed but apparent realignment of proxy and true objectives. We leave detailed investigation of reward grokking and its implications for hedging strategies to future work.

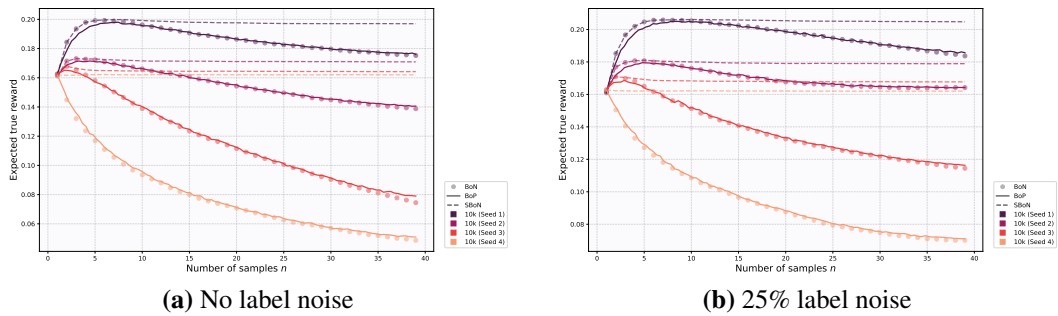

**(a)** No label noise          **(b)** 25% label noise

Figure 9: Expected true-reward vs. average number of samples with proxy trained on 10 000 examples: (a) without label noise; (b) with 25% label noise.

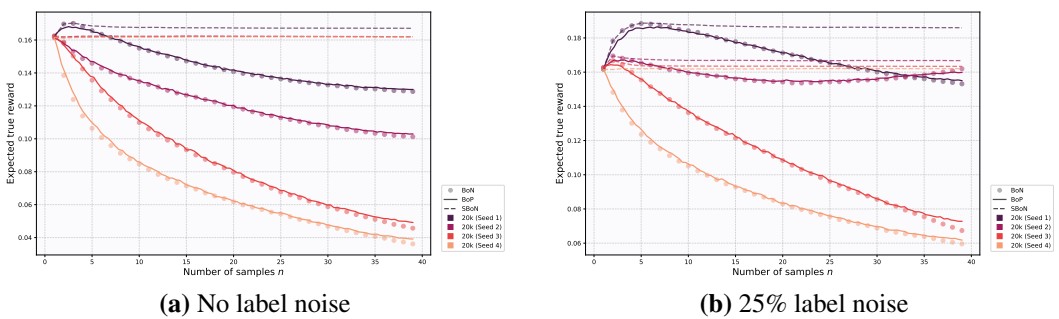

**(a)** No label noise          **(b)** 25% label noise

Figure 10: Expected true-reward vs. average number of samples with proxy trained on 20 000 examples: (a) without label noise; (b) with 25% label noise.

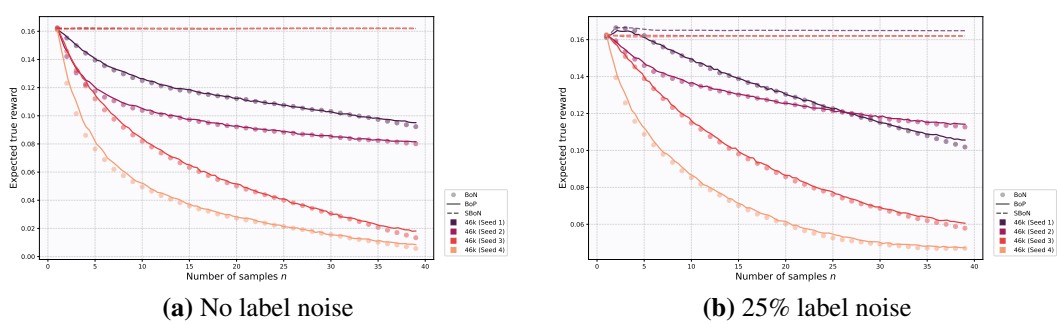

**(a)** No label noise          **(b)** 25% label noise

Figure 11: Expected true-reward vs. average number of samples with proxy trained on 46 000 examples: (a) without label noise; (b) with 25% label noise.

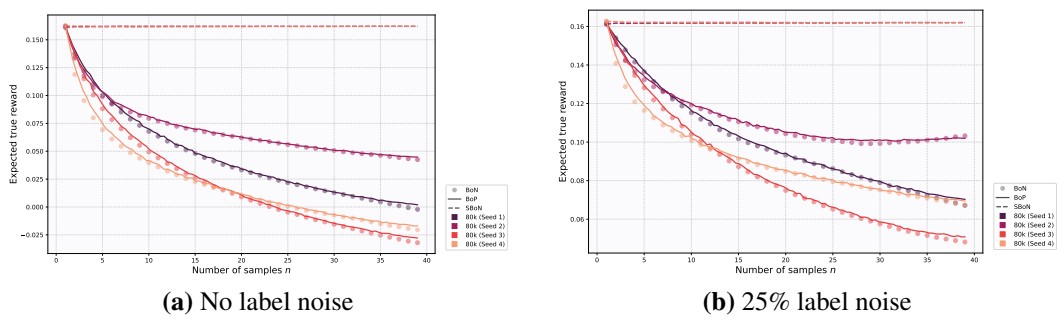

**(a)** No label noise          **(b)** 25% label noise

Figure 12: Expected true-reward vs. average number of samples with proxy trained on 80 000 examples: (a) without label noise; (b) with 25% label noise.

