# OpenReview forum: "Inference-Time Reward Hacking in Large Language Models"
_NeurIPS.cc/2025/Conference — NeurIPS 2025 spotlight_

### Official Review · Reviewer_Eaod · 2025-06-12

**Clarity:** 3
**Significance:** 3
**Originality:** 4
**Rating:** 4
**Confidence:** 3

**Summary:**

This paper addresses the fundamental challenge of reward hacking in LLM alignment, particularly focusing on inference-time methods like BoN sampling. The authors mathematically characterize reward hacking, proving its inevitability under common conditions where proxy rewards are imperfect approximations of true rewards. To mitigate this, they propose "hedging" strategies and introduce a novel inference-time method called Best-of-Poisson (BoP), which aims to approximate the optimal reward-tilted distribution with a single parameter. They also develop HedgeTune, an algorithm for finding the optimal hedging parameter across BoN, Soft Best-of-n (SBON), and BoP. Experimental results, both in synthetic and RLHF scenarios, demonstrate that hedging mitigates reward hacking and achieves superior distortion-reward tradeoffs.

**Questions:**

1. Can the framework be extended to work with noisy, approximated, or preference-based true rewards, and if so, what are the theoretical guarantees and practical implications?
1. What is the sensitivity of Theorem 1's inevitability claim and HedgeTune's performance to deviations from the assumption that proxy reward scores follow a uniform distribution? Have you explored the impact of non-uniform proxy reward distributions, and if so, what were the findings?
1. Given the identified inevitability of reward hacking, how do you foresee these hedging strategies scaling to much larger LLMs (e.g., 70B) and more complex, open-ended alignment tasks beyond the scope of the current experiments?

**Ethical Concerns:**

["NO or VERY MINOR ethics concerns only"]

**Final Justification:**

I have read the authors' detailed rebuttal and found it to be helpful. It has resolved the main concerns that made my initial assessment "borderline" and has reinforced my belief that this is a strong contribution. I am maintaining my positive score and raising the confidence score to 3.

**Limitations:**

Yes

**Paper Formatting Concerns:**

No major formatting concerns were observed. The paper generally adheres to the NeurIPS format. But there appears to be a missing line number between lines 193 and 194.

**Quality:**

3

**Strengths And Weaknesses:**

**Strengths**

1. **Novelty in Hedging for Reward Hacking:** The paper introduces a principled approach to "hedging" against reward hacking specifically at inference time, offering a new perspective on mitigating this pervasive problem.
1. **Theoretical Characterization:** The formalization and proof of the inevitability of reward hacking (Theorem 1) provide a strong theoretical foundation for the observed empirical phenomenon, which is crucial for understanding its mechanisms.
1. **Best-of-Poisson (BoP) as a Novel Method:** BoP is presented as a mathematically elegant and computationally efficient alternative to BoN, capable of approximating optimal reward-distortion tradeoffs with a single tunable parameter, which simplifies practical application.
1. **Practical Algorithm (HedgeTune):** The HedgeTune algorithm offers a concrete method for operationalizing the hedging strategy by identifying the optimal parameter to balance reward exploitation and fidelity to the reference distribution.

**Weaknesses**

1. **Practicality of True Reward Oracle Requirement:** A limitation is the explicit requirement for "access to the true reward" (Section 4, line 234) for HedgeTune to operate. In real-world LLM alignment, the true reward is often unknown, costly, or impossible to precisely quantify. If the true reward were readily available, simpler pre-experiment hyperparameter tuning could potentially find the optimal operating point without needing sophisticated hedging algorithms, thus diminishing the practical utility of the proposed method in its current form. This assumption significantly limits the immediate applicability of HedgeTune.
1. **Limited Justification for Proxy Uniformity Assumption:** The paper assumes that proxy reward scores have a uniform distribution (Section 2, line 120-122; Section 4, line 236). While stated as incurring "little loss in generality," the impact of this strong assumption on the theoretical guarantees and practical performance, especially if real-world proxy reward distributions deviate significantly from uniform, is not thoroughly explored.
1. **Experimental Scope and Real-World Applicability:** While experiments include a "hiding in the wild" setup, the overall experimental validation might still be limited. The use of Pythia-1.4B models and a synthetic reward hacking scenario, while controlled, may not fully capture the complexities and nuances of reward hacking observed in larger, more diverse LLMs and real-world alignment tasks.

---

> ### Author Rebuttal · Authors · 2025-07-31
>
> Thank you for your feedback! We will address each of your points below.
>
> ---
>
> # W1: *Practicality of True Reward Oracle Requirement*
>
> You've highlighted a key practical challenge: the high cost of true reward data. HedgeTune is designed for this exact scenario. Instead of a costly brute-force search, it leverages the unimodality guaranteed by Theorem 1 to apply an efficient root-finding algorithm. This finds the optimal parameter with significantly fewer expensive validation samples. Our sample complexity analysis, detailed in our response to Reviewer pDYx, provides a formal treatment of this efficiency. We will add a discussion on the practical limitation to the conclusion.
>
> ---
>
> # Q1: *Working with noisy, approximated, or preference-based true rewards*
>
> This is an excellent question about future directions, which we will add to our conclusion.
>
> **Noisy/Approximated Rewards:** We hypothesize that the true reward's unimodal structure acts as a natural regularizer, helping HedgeTune find the right peak. However, more work is needed for a formal theoretical treatment.
>
> **Preference-Based Rewards:** This is the most exciting and challenging future direction. One promising method would be to design a surrogate objective based on preference data akin to DPO. A central theoretical question would be to prove that this new objective also exhibits a unimodal or single-crossing property. We believe this is a rich area for future work. We will add a detailed discussion of these potential extensions to our conclusion.
>
> ---
>
> # W2/Q2: *Proxy Uniformity Assumption*
>
> Thank you for these points. **The proxy reward scores do not need to be uniform themselves.**
>
> We first map the proxy rewards to a standardized discrete space by transforming them into their quantiles using their own empirical CDF $F_p$. This process, defined by the relation
> $$ u = F_p(r_p) $$
> ensures that the new reward $u$ is uniform on $[0, 1]$ by construction. This transformation preserves the rank-ordering of the scores: an output with a higher proxy reward will also have a higher transformed score. Of course -- the resulting random variable will not be a continuous uniform random variable (which we assume), and instead will be a discrete uniform random variable. Recent work [1] has shown for a dense enough set of examples, the error incurred by this continuous model is negligible in the context of LLMs. We recognize this point was not sufficiently clear in the manuscript, and we will revise it to precisely explain our construction.
>
> Nevertheless, your question correctly highlights if this approximation step could be entirely removed. **Theorem 1 can in fact apply directly for the case where rewards have discrete support** with small changes to the current proof (swapping integrals with sums). This is particularly exciting, as it confirms that reward hacking follows a predictable behavior and this is a **fundamental** property of these inference methods.
>
> Instead of the continuous reward $u \in [0, 1]$, we consider a discrete set of $M$ possible outputs, ranked from worst to best according to their proxy reward scores. Let the true reward for the output at rank $k$ be $w_k$. The probability matrix $P(\theta, k) = p_k(\theta)$ gives the probability of selecting the output at rank $k$ given an inference-time parameter $\theta$. The expected true reward is now expressed as a discrete sum:
> $$
> \mathbb{E}[R_t \mid \theta] = \sum_{k=1}^M p_k(\theta) \cdot w_k
> $$
> We note that the *Variation-Diminishing Theorem* we employ is a general result that applies to discrete sums just as it does to continuous integrals. The Variation-Diminishing Theorem guarantees that if our probability matrix $P$ is $TP_2$, then expected true reward $\mathbb{E}[R_t \mid \theta]$ remains simple and unimodal. The core result—and thus our ability to find a single optimal HedgeTune parameter—holds! This restated proof solidifies our findings and will be incorporated into the final paper. We thank you for prompting this important extension.
>
> ---
>
> # W3: *Experimental Scope and Real-World Applicability*
>
> Thank you for this great question. To address this, we ran complementary experiments showing our method mitigates reward hacking on complex reasoning benchmarks (MMLU Pro, MATH, GPQA) even with large 8B proxy reward models!
>
> 1.  *Datasets:* MMLU Pro (complex reasoning), MATH (mathematical problem solving), and GPQA (questions in natural sciences). A correct response gets a true reward of 1, while an incorrect response gets a true reward of 0.
> 2.  *LLMs:* We leverage an open-source dataset with multiple responses generated per benchmark question from models such as GPT-4o-mini and Claude Haiku 3 [2].
> 3.  *Proxy Rewards:* Each response is scored using various reward models, including InterLM-2 **1.8B**, Llama-3-Offset-Bias **8B**, and Skywork-Llama-3.1 **8B** [2].
>
> The authors of [2] note reward hacking with Best-of-$n$ on a suite of reward models and datasets. Due to time constraints during the rebuttal period, we only test Soft Best-of-$n$; however, we intend to show the results for all the methods in the final paper. We present the results in the table below:
>
> | $n$ | MMLU (Llama-3 8B) BoN | MMLU (Llama-3 8B) Soft BoN | MMLU (internlm2-1.8b) BoN | MMLU (internlm2-1.8b) Soft BoN | GPQA (Skywork-Llama 3.1 8B) BoN | GPQA (Skywork-Llama 3.1 8B) Soft BoN | MATH (internlm2-1.8b) BoN | MATH (internlm2-1.8b) Soft BoN |
> |:---:|:---:|:---:|:---:|:---:|:---:|:---:|:---:|:---:|
> | 1 | 47.51±1.08 | 47.51±1.08 | 47.57±1.08 | 47.57±1.08 | 40.70±1.02 | 40.70±1.02 | 34.38±1.10 | 34.38±1.10 |
> | 2 | 53.10±1.24 | 53.12±1.24 | 50.33±1.18 | 50.32±1.18 | 42.98±1.17 | 42.98±1.16 | 37.32±1.18 | 37.34±1.18 |
> | 4 | 56.23±1.40 | 56.30±1.40 | 51.38±1.32 | 51.37±1.32 | 44.44±1.31 | 44.42±1.31 | 39.01±1.28 | 39.04±1.28 |
> | 6 | 57.21±1.50 | 57.32±1.49 | 51.78±1.41 | 51.75±1.41 | 44.98±1.40 | 44.95±1.38 | **39.38±1.34** | **39.39±1.33** |
> | 10 | **57.67±1.63** | 57.92±1.58 | 52.01±1.55 | 51.99±1.54 | 45.35±1.53 | **45.25±1.52** | 39.19±1.44 | 39.31±1.40 |
> | 12 | 57.65±1.68 | **58.01±1.65** | **52.12±1.60** | **52.13±1.60** | **45.36±1.58** | 45.22±1.53 | 39.03±1.50 | 39.24±1.42 |
> | 14 | 57.49±1.73 | 57.96±1.66 | 52.08±1.66 | 52.06±1.65 | 45.26±1.63 | 45.20±1.55 | 38.81±1.55 | 39.10±1.48 |
> | 18 | 57.13±1.82 | 57.98±1.72 | 52.01±1.76 | 52.03±1.75 | 44.89±1.74 | 45.04±1.53 | 38.25±1.65 | 38.97±1.49 |
> | 22 | 56.65±1.92 | 57.89±1.72 | 51.81±1.87 | 51.80±1.85 | 44.46±1.85 | 45.01±1.53 | 37.77±1.77 | 38.83±1.51 |
> | 26 | 56.34±2.01 | 57.90±1.75 | 51.38±1.99 | 51.49±1.64 | 43.98±1.97 | 44.92±1.53 | 37.34±1.90 | 38.72±1.43 |
> | 30 | 56.10±2.13 | 57.92±1.79 | 50.82±2.13 | 51.38±1.75 | 43.59±2.11 | 44.90±1.65 | 37.06±2.05 | 38.66±1.45 |
> | 32 | 56.05±2.19 | 57.94±1.81 | 50.48±2.21 | 51.29±1.77 | 43.35±2.19 | 44.86±1.62 | 36.91±2.13 | 38.66±1.51 |
>
>
> **The observed reward hacking curve exactly matches our theoretical prediction in Theorem 1.** For example, Best-of-$n$ clearly demonstrates hacking on the GPQA dataset, even with the very capable Skywork-Llama 3.1 8B as proxy reward (ranked # 10 out of all non-generative reward models on RewardBench [3]). While performance initially climbs from 40.70% at $n = 1$ to a peak of 45.36% at $n = 12$, it ultimately decreases to 43.35% as over-optimization sets in.
>
> **Hedging mitigates reward hacking in all cases.** Soft BoN delivers more stable and often superior performance than BoN. On GPQA, it reduces the performance drop from over 2 points to less than 0.4 points. On MATH, where hacking is most severe, it effectively mitigates BoN’s 2.5-point collapse. This stability also enables Soft BoN to achieve a higher peak score on MMLU (Llama-3 8B).
>
> The hedging benefit holds for learning from human preferences (RLHF), complex reasoning (MMLU), math (MATH), and science questions (GPQA), and is demonstrated with multiple large reward models (8B parameters). Please let us know if you have any additional questions about our setup or our results!
>
> ---
>
> # Q3: *Scaling to larger LLMs and complex, open-ended tasks*
>
> We hope that our supplementary experimental results on complex reasoning demonstrate the effectiveness of hedging, even with larger reward models. The advantage of hedging is that it *does not change* across setups. Given access to "true" reward measurements, you would tune inference-time parameters to operate in a regime where you do not over-optimize to a proxy reward.
>
> What does change is our ability to observe differences between true and proxy rewards, or even obtain such measurements. This might be feasible in narrow tasks, but more difficult in LLMs that are used across tasks like summarizing financial documents or refactoring a legacy codebase. Obtaining a consistent and reliable "true reward" signal in such domains is a significant research challenge in itself. If a form of hacking is undetectable by our evaluation process (human or otherwise), HedgeTune cannot guard against it. As the research community develops better methods for measuring hacking, we view HedgeTune as an immediate, lightweight, and principled tool to translate that into a more robust inference-time policy.
>
>
> ---
>
> # Summary:
>
> We thank you for the thoughtful questions that have significantly clarified and strengthened our work. In response to your feedback, we will reframe HedgeTune's practicality for real-world tasks with expensive but limited ground-truth data. We have also extended our core theory to the discrete case and demonstrated our method's scalability with extensive new experiments on reasoning benchmarks. We hope these additions directly address your concerns and we appreciate your help in improving the paper.
>
> ---
>
> **References:**
> 1.  Gui et al. *BoNBoN Alignment for Large Language Models and the Sweetness of Best-of-n Sampling*. NeurIPS 2024.
> 2.  Frick et al. *How to Evaluate Reward Models for RLHF*. ICLR 2025.
> 3.  Lambert et al. "RewardBench: Evaluating Reward Models for Language Modeling". NAACL 2025.

---

> > ### Comment · Reviewer_Eaod · 2025-08-05
> >
> > Thank you for the detailed and insightful rebuttal. These additions have substantially strengthened the paper and solidified my positive assessment. I am maintaining my recommendation for acceptance.

---

### Official Review · Reviewer_51Rg · 2025-06-24

**Clarity:** 3
**Significance:** 2
**Originality:** 2
**Rating:** 4
**Confidence:** 3

**Summary:**

This paper addresses the problem of reward hacking in large language models (LLMs) that arises from overoptimizing imperfect proxy reward models during inference-time alignment. The authors formalize this phenomenon and introduce the concept of the "hacking threshold", the point beyond which optimizing for proxy rewards degrades true performance. They propose a new inference-time method, Best-of-Poisson (BoP), which samples a variable number of outputs based on a Poisson distribution to balance reward and distortion with a single parameter. They also introduce HedgeTune, an algorithm for tuning these methods to optimize true reward while avoiding overoptimization. Experiments demonstrate the benefits of hedging in improving distortion-reward tradeoffs.

**Questions:**

1. Theorem 1 relies on technical assumptions such as $\textnormal{TP}_2$ and strictly MLR. For readers without a precise technical background, could the authors provide additional intuition or interpretation for these assumptions? In particular, clarifying how these conditions correspond to realistic properties of models used in practice would help make the theoretical contributions more accessible and broadly relevant.

2. The reviewer finds the empirical evaluation somewhat limited in scope, which makes it difficult to assess the generalizability of the proposed methods across different LLM architectures, datasets, and proxy reward models. While the use of true rewards is valuable for controlled analysis, it would also be helpful to evaluate performance on established alignment benchmarks that provide scores in terms of the alignment of LLMs. Such benchmarks could offer complementary evidence for the effectiveness of hedging strategies in more realistic or diverse settings.

3. How does the proposed hedging method compare with other approaches for mitigating reward hacking, such as regularization, ensembling, etc.? A comparative empirical or conceptual analysis against these baselines would help situate the proposed approach more clearly within the broader literature.

I'm happy to consider increasing my score if the authors provide compelling responses to my questions, particularly by clarifying theoretical assumptions and expanding the empirical evidence to support the general applicability of their methods.

**Ethical Concerns:**

["NO or VERY MINOR ethics concerns only"]

**Final Justification:**

Based on the rebuttal of the authors with the additional experiments, the reviewer finds the empirical evaluation more convincing. Therefore, my recommended score increases to 4: Borderline accept.

**Limitations:**

yes

**Quality:**

2

**Strengths And Weaknesses:**

Strengths: This paper provides a rigorous analysis of reward hacking in inference-time alignment for language models, combining theoretical insights with practical methods. The presentation is generally clear, with key concepts like the hacking threshold and hedging strategies explained in a comprehensible way. The work addresses an important problem in LLM alignment and contributes useful tools, such as Best-of-Poisson (with the advantage of having a single tunable parameter) and HedgeTune, that offer practical value. While the core ideas build on existing alignment techniques, the formulation and integration of hedging into inference-time methods add a degree of originality and relevance.

Weaknesses: The presentation of the paper could be improved in the sense that several figures are difficult to read due to formatting and some passages would benefit from proofreading to correct minor typos and enhance readability. Although the paper addresses a relevant problem and proposes useful methods, the empirical evaluation is relatively narrow and may not fully demonstrate the generality of the approach.

---

> ### Author Rebuttal · Authors · 2025-07-31
>
> Thank you for the thoughtful review. We address your questions and comments below and would be happy to answer any additional questions during the discussion period.
>
> ---
>
> # Q1: *Additional intuition on technical assumptions like $TP_2$ and strictly MLR*
>
> We will add the following discussion about the technical conditions to the paper.
>
> **What the Technical Conditions Mean Intuitively:**
> The conditions guarantee that the inference-time policies behave in a *well-ordered* manner: increasing the tuning parameter $\theta$ consistently makes the policy "greedier" and more likely to select outputs with high proxy rewards. The clearest example is the $n$ in Best-of-$n$: the larger the $n$ (which is generalized by our $\theta$ parameter), the more aggressively we optimize for the proxy.
>
> This provides the most natural setting to study reward hacking. The monotone likelihood ratio (MLR) property induces a simple pairwise ordering: for any two policies with different parameters, the policy with the larger parameter is greedier. This property arises as a consequence of the stronger total positivity of order 2 ($TP_2$) condition. $TP_2$ is a structural property of the entire policy family, ensuring that such monotonic behavior holds uniformly across all parameter values and outcomes. It is a standard tool in the study of order statistics, utility theory, and stochastic processes.
>
> **Why They Help Us Understand Hacking:**
> The core challenge of reward hacking is that the true reward function can be complex. This is where our theoretical contribution yields a significant practical payoff. We leverage the Variation-Diminishing Theorem to show that for **any** bounded reward, the expected true reward must follow a simple, predictable path that we characterize exactly. This result makes the problem of reward hacking tractable and allows us to search for the single, optimal *hacking threshold*.
>
> ---
>
> # W2/Q2: *Empirical Setup*
>
> Thank you for this great question. To address this, we ran complementary experiments showing **our method mitigates reward hacking on complex reasoning benchmarks** (MMLU Pro, MATH, GPQA) even with large 8B proxy reward models.
>
> 1.  *Datasets:* MMLU Pro (complex reasoning), MATH (mathematical problem solving), and GPQA (questions in natural sciences). A correct response gets a true reward of 1, while an incorrect response gets a true reward of 0.
> 2.  *LLMs:* We leverage an open-source dataset with multiple responses generated per benchmark question from models such as GPT-4o-mini and Claude Haiku 3 [1].
> 3.  *Proxy Rewards:* Each response is scored using various reward models, including InterLM-2 **1.8B**, Llama-3-Offset-Bias **8B**, and Skywork-Llama-3.1 **8B** [1].
>
> The authors of [1] note reward hacking with Best-of-$n$ on a suite of reward models and datasets. Due to time constraints during the rebuttal period, we only test Soft Best-of-$n$; however, we intend to show the results for all the methods in the final paper. We present the results in the table below:
>
> | $n$ | MMLU (Llama-3 8B) BoN | MMLU (Llama-3 8B) Soft BoN | MMLU (internlm2-1.8b) BoN | MMLU (internlm2-1.8b) Soft BoN | GPQA (Skywork-Llama 3.1 8B) BoN | GPQA (Skywork-Llama 3.1 8B) Soft BoN | MATH (internlm2-1.8b) BoN | MATH (internlm2-1.8b) Soft BoN |
> |:---:|:---:|:---:|:---:|:---:|:---:|:---:|:---:|:---:|
> | 1 | 47.51±1.08 | 47.51±1.08 | 47.57±1.08 | 47.57±1.08 | 40.70±1.02 | 40.70±1.02 | 34.38±1.10 | 34.38±1.10 |
> | 2 | 53.10±1.24 | 53.12±1.24 | 50.33±1.18 | 50.32±1.18 | 42.98±1.17 | 42.98±1.16 | 37.32±1.18 | 37.34±1.18 |
> | 4 | 56.23±1.40 | 56.30±1.40 | 51.38±1.32 | 51.37±1.32 | 44.44±1.31 | 44.42±1.31 | 39.01±1.28 | 39.04±1.28 |
> | 6 | 57.21±1.50 | 57.32±1.49 | 51.78±1.41 | 51.75±1.41 | 44.98±1.40 | 44.95±1.38 | **39.38±1.34** | **39.39±1.33** |
> | 10 | **57.67±1.63** | 57.92±1.58 | 52.01±1.55 | 51.99±1.54 | 45.35±1.53 | **45.25±1.52** | 39.19±1.44 | 39.31±1.40 |
> | 12 | 57.65±1.68 | **58.01±1.65** | **52.12±1.60** | **52.13±1.60** | **45.36±1.58** | 45.22±1.53 | 39.03±1.50 | 39.24±1.42 |
> | 14 | 57.49±1.73 | 57.96±1.66 | 52.08±1.66 | 52.06±1.65 | 45.26±1.63 | 45.20±1.55 | 38.81±1.55 | 39.10±1.48 |
> | 18 | 57.13±1.82 | 57.98±1.72 | 52.01±1.76 | 52.03±1.75 | 44.89±1.74 | 45.04±1.53 | 38.25±1.65 | 38.97±1.49 |
> | 22 | 56.65±1.92 | 57.89±1.72 | 51.81±1.87 | 51.80±1.85 | 44.46±1.85 | 45.01±1.53 | 37.77±1.77 | 38.83±1.51 |
> | 26 | 56.34±2.01 | 57.90±1.75 | 51.38±1.99 | 51.49±1.64 | 43.98±1.97 | 44.92±1.53 | 37.34±1.90 | 38.72±1.43 |
> | 30 | 56.10±2.13 | 57.92±1.79 | 50.82±2.13 | 51.38±1.75 | 43.59±2.11 | 44.90±1.65 | 37.06±2.05 | 38.66±1.45 |
> | 32 | 56.05±2.19 | 57.94±1.81 | 50.48±2.21 | 51.29±1.77 | 43.35±2.19 | 44.86±1.62 | 36.91±2.13 | 38.66±1.51 |
>
> **The observed reward hacking curve exactly matches our theoretical prediction in Theorem 1.** For example, Best-of-$n$ clearly demonstrates hacking on the GPQA dataset, even with the very capable Skywork-Llama 3.1 8B as proxy reward (ranked #10 out of all non-generative reward models on RewardBench [2]). While performance initially climbs from $40.70\%$ at $n = 1$ to a peak of $45.36\%$ at $n = 12$, it ultimately decreases to $43.35\%$ as over-optimization sets in.
>
> **Hedging mitigates reward hacking in all cases.** Soft BoN delivers more stable and often superior performance than BoN. On GPQA, it reduces the performance drop from over 2 points to less than 0.4 points. On MATH, where hacking is most severe, it effectively mitigates BoN’s 2.5-point collapse. This stability also enables Soft BoN to achieve a higher peak score on MMLU (Llama-3 8B).
>
> The hedging benefit holds for learning from human preferences (RLHF), complex reasoning (MMLU), math (MATH), and science questions (GPQA), and is demonstrated with multiple large reward models (8B parameters). Please let us know if you have any additional questions about our setup or our results.
>
> ### References:
> 1.  Frick et al. *How to Evaluate Reward Models for RLHF*. ICLR 2025.
> 2.  Lambert et al. *RewardBench: Evaluating Reward Models for Language Modeling*. NAACL 2025.
>
> ---
>
> # Q3: ***How does hedging compare with other approaches for mitigating reward hacking***
>
> Thank you for this excellent question. We provide the following conceptual analysis. The research community has developed several important strategies to combat reward hacking, which (broadly speaking) fall into the following categories:
>
> **Training-Time Regularization.**
> This is the most common approach, typified by adding a KL-divergence penalty to the reward maximization objective as in RLHF. The goal is to prevent the policy from deviating too far from a trusted reference model. While a vital first line of defense, this approach requires expensive full model retraining and risks under-optimization if the penalty is too strong.
>
> **Improving the Proxy Reward Model.**
> A second category aims to build a more robust reward signal that is inherently harder to hack. Ensembling involves averaging scores from multiple, independently trained reward models. However, this method multiplies inference costs and cannot fix systemic biases shared by all models in the ensemble. Meanwhile, robust training methods (e.g., RRM, InfoRM) modify the reward model's training process itself to disentangle response quality from spurious cues, introducing significant complexity to the training pipeline.
>
> **Inference-Time Methods.**
> The closest analogues to our work include:
> *   **Regularized Best-of-$n$ (RBoN):** A powerful heuristic that balances the proxy reward against a penalty term: $R(x) - \beta \cdot \text{Penalty}(x)$.
> *   **Inference-Time Pessimism:** This method uses a principled rejection sampling step, which requires estimating a normalization constant.
>
> **Our Framework:**
> Hedging introduces a distinct approach that operates purely at inference time. It offers two key advantages:
>
> 1.  **Zero Retraining Cost and Maximum Flexibility:** Hedging is applied to an existing, trained policy using existing proxy reward models. A practitioner can take a single policy and calibrate it against different proxy rewards or for different tasks without incurring any retraining costs.
> 2.  **Principled and Efficient:** We prove (Theorem 1) that the expected true reward function follows a predictable "rise-and-fall" path with a single optimal peak. HedgeTune therefore replaces heuristic balancing or more involved sampling schemes with a principled search for this unique optimum—a problem we prove is tractable.
>
> However, one key limitation is that hedging requires a tuning dataset with "ground truth" labels. Ultimately, we see hedging not as a replacement for some of the previous methods. Instead, HedgeTune offers a computationally lightweight, theoretically-grounded, and highly flexible method to mitigate hacking at deployment.
>
> ---
>
> # W1: *The presentation of the paper could be improved*
>
> **Answer:** We have gone through the paper again to fix typos and improve figure formatting. The major edits include enhancing the figure legends and style, streamlining the discussion of technical assumptions in Section 2 (e.g., merging Lines 153-160 with Lines 177-185), providing more direct proofs in the appendix, and making other minor edits throughout the paper. We believe these changes significantly improve the paper's quality and appreciate your helpful feedback.
>
> ---
>
> # Summary:
> We thank you for providing a clear path to improve the paper. In response, we have demonstrated our method’s practical generality with new experiments on complex reasoning benchmarks (MMLU, MATH, GPQA) using large reward models. We have also clarified the intuition behind our core theoretical assumptions to improve accessibility, situated our work with a new conceptual analysis of alternative mitigation methods, and improved the presentation of the paper. We believe the paper is significantly stronger as a result of your review, so thank you!

---

> > ### Comment · Reviewer_51Rg · 2025-08-04
> > **Response to the rebuttal**
> >
> > I thank the authors for their thoughtful and detailed rebuttal. I have carefully reviewed their responses and the additional experiments they provided. They are certainly helpful and I will re-evaluate the paper accordingly.
> >
> > Definitely not asking for a new experiment, but just asking out of curiosity. Why did the authors choose reasoning benchmarks for additional experiments instead of alignment benchmarks? Is it not the case that reward hacking in general is most severe in instruction-following tasks? I feel like here in these challenging reasoning benchmarks, reward hacking does not seem to be that severe. Just asking to understand better...

---

> > > ### Author Response · Authors · 2025-08-04
> > >
> > > We chose reasoning benchmarks to address your question on the generalizability of our results (Q2). Reasoning benchmarks offer a clear, objective "true reward" (correct vs. incorrect). Consequently, they provide a verifiable setting that complements the human-preference benchmarks already in our paper.
> > >
> > > Thanks again for your feedback and support!

---

> > > > ### Comment · Reviewer_51Rg · 2025-08-04
> > > > **Response to the authors' comment**
> > > >
> > > > I understand, thanks very much for responding.

---

### Official Review · Reviewer_pDYx · 2025-07-01

**Clarity:** 3
**Significance:** 3
**Originality:** 3
**Rating:** 4
**Confidence:** 4

**Summary:**

This paper studies the phenomenon of inference-time reward hacking when LLMs are steered by imperfect proxy reward models.  The authors (1) prove in Theorem 1 that, under mild TP2/MLR assumptions, the true reward as a function of an inference-time tuning parameter must either increase monotonically or exhibit a single peak before collapsing; (2) introduce Best-of-Poisson (BoP) sampling, which uses a Poisson-distributed sample count to approximate the ideal KL-constrained tilted distribution with a single parameter; and (3) propose HedgeTune, a root-finding algorithm that selects the optimal inference-time parameter (for BoN, SBoN, or BoP) to hedge against over-optimization, demonstrating in both synthetic and RLHF-style experiments that hedging achieves superior true-reward vs. distortion trade-offs with minimal overhead.

**Questions:**

1. **Sample efficiency:** How many true-reward evaluations does HedgeTune require to reliably converge to the optimal parameter?

2. **Task breadth:** How does HedgeTune generalize to more complex reasoning tasks where a ground-truth reward is available?

**Minor issue:**

- Figure 3’s y-axis is mislabeled. It reads “Reward Improvement over Base Distribution (%)” but the values are shown as decimals (e.g., 0.8–0.9) rather than true percentages.

**Ethical Concerns:**

["NO or VERY MINOR ethics concerns only"]

**Final Justification:**

I have read all of the reviews and author responses. My moderate assessment of 4 is consistent with most of the reviews and reviewers.

**Limitations:**

Yes

**Paper Formatting Concerns:**

None noted.

**Quality:**

3

**Strengths And Weaknesses:**

## Strengths
- The paper tackles the intriguing problem of inference-time reward hacking, precisely defines it, and offers rigorous theoretical analysis.
- Using controlled synthetic setups and realistic LLM + proxy-RM pipelines show marked improvement in true-reward by hedging.

## Weaknesses
The key point of this paper's method is : **You must have a set of samples for which you can measure the true reward** (e.g., a validation set), or an equivalent human evaluator / high-quality scoring function, so that you can estimate the true reward curve. You then use a root-finding method to locate the peak where $g^{\prime}(\vartheta)=0$ .

Without a “true-reward evaluation set” or an unbiased, high-quality evaluator, HedgeTune cannot locate the true peak and thus cannot automatically select the optimal inference-time parameter. This limitation is particularly acute for datasets without an objective ground truth. As a result, HedgeTune may become impractical for many real-world tasks.

---

> ### Author Rebuttal · Authors · 2025-07-31
>
> We sincerely thank you for your detailed and thoughtful feedback. We address next the questions and weaknesses raised in your review.
>
>  ---
>
> # Q1: ***Sample efficiency of HedgeTune***
> We agree sample efficiency is an important addition. We provide a new formal analysis, which we will add to the paper. As you noted, HedgeTune finds the root of an empirical gradient constructed from a dataset of $M$ true-reward evaluations. We leverage our paper's key theoretical contribution: under Theorem 1, the true gradient $R(\theta)$ will have **exactly one-root**, moving from positive to negative. Under boundedness assumptions, we can apply standard concentration inequalities like Hoeffding's inequality.
>
> We remind the reviewer of the following expression for the gradient of the true reward (see Eq. 8 in Appendix A for the derivation):
>
> $$R(\theta) = E_{X \sim p_\theta}[r_t(X) \cdot  \varphi(X, \theta)] = E_{U \sim \text{Unif}[0,1]}[r_t(U) \cdot  \varphi(U, \theta) \cdot p_\theta(U)] = E_{U \sim \text{Unif}[0,1]}[h(U, \theta)]$$
>
> where $r_t(\cdot)$ is the true reward function, $\varphi(\cdot, \cdot)$ is the score function, and $p_\theta$ is the density of the inference-time policy parameterized by $\theta$. We consider the following assumptions on a compact parameter space $\Theta = [\theta_{\min}, \theta_{\max}]$ (equivalent to the search range of our algorithm):
>
> **Assumption 1 (Unimodality)**
> The true performance function $f(\theta) = E_{X \sim p_\theta}[r_t(X)]$ is unimodal with a unique maximizer $\theta^* \in \Theta$, so its derivative, $R(\theta)$, is monotonically decreasing. This holds under Theorem 1.
>
> **Assumption 2 (Bounded Gradient Summands)**
> There exists a constant $H < \infty$ such that:
> $$
> |h(u, \theta)| \le H \quad \text{for all } u \in [0,1] \text{ and } \theta \in \Theta
> $$
>
> **Assumption 3 (Strong Slope)**
> The true gradient $R(\theta)$ has a non-zero slope at its root $\theta^\ast$. That is, there exists a constant $m>0$ such that for all $\theta \in \Theta$:
> $$
> |R(\theta)| \ge m|\theta - \theta^\ast|
> $$
>
> ## Theorem
> To guarantee that the estimated optimum $\theta^\dagger$ is within a desired accuracy $\epsilon$ of the true optimum $\theta^\ast$ with probability at least $1 - \alpha$, the required number of true-reward evaluations $M$ must satisfy:
> $$
> M = \mathcal{O}\left( \frac{H^2}{m^2 \epsilon^2} \log\left(\frac{1}{\alpha}\right) \right)
> $$
>
> **Proof**
>
> The proof relies on showing that for a sufficiently large $M$, the empirical gradient will, with high probability, have opposite signs at the boundaries of the interval $[\theta^\ast - \epsilon, \theta^\ast + \epsilon]$. By the Strong Slope assumption (A3), the true gradient $R(\theta)$ is bounded away from zero at the points $\theta_{lo} = \theta^\ast - \epsilon$ and $\theta_{hi} = \theta^\ast + \epsilon$:
> *   At $\theta_{hi}$: $R(\theta_{hi}) \le -m|\theta_{hi} - \theta^*| = -m\epsilon$.
> *   At $\theta_{lo}$: $R(\theta_{lo}) \ge m|\theta_{lo} - \theta^*| = m\epsilon$.
>
> Let $\mathcal{F}$ be the event that the empirical gradient ${\hat{R}}(\theta)$  does not respect these signs, then by Hoeffding's inequality, we have that:
>
> $$\begin{aligned}
> \Pr[\mathcal{F}] &= \Pr\left[ ({\hat{R}}(\theta_{hi}) \ge 0) \lor ({\hat{R}}(\theta_{lo}) \le 0) \right] \\\\
> &\le \Pr[{\hat{R}}(\theta_{hi}) - R(\theta_{hi}) \ge m\epsilon] + \Pr[{\hat{R}}(\theta_{lo}) - R(\theta_{lo}) \le -m\epsilon] \quad \text{(by Union Bound and A3)} \\\\
> &\le \exp\left(-\frac{M(m\epsilon)^2}{2H^2}\right) + \exp\left(-\frac{M(m\epsilon)^2}{2H^2}\right) = 2\exp\left(-\frac{M m^2 \epsilon^2}{2H^2}\right)
> \end{aligned}$$
>
> To guarantee a confidence level of $1-\alpha$, we require that:
> $$
> M \ge \frac{2H^2}{m^2\epsilon^2}\log\left(\frac{2}{\alpha}\right)
> $$
> Then, with probability at least $1-\alpha$, we have $\hat{g}_M(\theta^* - \epsilon) > 0$ and $\hat{g}_M(\theta^* + \epsilon) < 0$. Since $\hat{g}_M(\theta)$ is a continuous function of $\theta$, the Intermediate Value Theorem guarantees the existence of a root $\theta^\dagger \in (\theta^* - \epsilon, \theta^* + \epsilon)$.
>
> The values $H^2$ (estimator variance) and $1/m^2$ (flatness of $\theta^\ast$) capture the problem's intrinsic difficulty. We will add this analysis to the paper, as well as explicit bounds for Best-of-$n$ and Best-of-Poisson. We would be happy to answer any follow-up question on this point.
>
> ---
>
> # Q2: ***Hedging in complex reasoning tasks where a ground-truth reward is available***
>
> Thank you for this great question. To address this, we ran complementary experiments showing *our method mitigates reward hacking on complex reasoning benchmarks* (MMLU Pro, MATH, GPQA) even with large 8B proxy reward models!
>
> 1.  *Datasets:* MMLU Pro (complex reasoning), MATH (mathematical problem solving), and GPQA (questions in natural sciences). A correct response gets a true reward of 1, while an incorrect response gets a true reward of 0.
> 2.  *LLMs:* We leverage an open-source dataset with multiple responses generated per benchmark question from models such as GPT-4o-mini and Claude Haiku 3 [1].
> 3.  *Proxy Rewards:* Each response is scored using various reward models, including InterLM-2 **1.8B**, Llama-3-Offset-Bias **8B**, and Skywork-Llama-3.1 **8B** [1].
>
> The authors of [1] note reward hacking with Best-of-$n$ on a suite of reward models and datasets. Due to time constraints during the rebuttal period, we only test Soft Best-of-$n$; however, we intend to show the results for all the methods in the final paper. We present the results in the table below:
>
> | $n$ | MMLU (Llama-3 8B) BoN | MMLU (Llama-3 8B) Soft BoN | MMLU (internlm2-1.8b) BoN | MMLU (internlm2-1.8b) Soft BoN | GPQA (Skywork-Llama 3.1 8B) BoN | GPQA (Skywork-Llama 3.1 8B) Soft BoN | MATH (internlm2-1.8b) BoN | MATH (internlm2-1.8b) Soft BoN |
> |:---:|:---:|:---:|:---:|:---:|:---:|:---:|:---:|:---:|
> | 1 | 47.51±1.08 | 47.51±1.08 | 47.57±1.08 | 47.57±1.08 | 40.70±1.02 | 40.70±1.02 | 34.38±1.10 | 34.38±1.10 |
> | 2 | 53.10±1.24 | 53.12±1.24 | 50.33±1.18 | 50.32±1.18 | 42.98±1.17 | 42.98±1.16 | 37.32±1.18 | 37.34±1.18 |
> | 4 | 56.23±1.40 | 56.30±1.40 | 51.38±1.32 | 51.37±1.32 | 44.44±1.31 | 44.42±1.31 | 39.01±1.28 | 39.04±1.28 |
> | 6 | 57.21±1.50 | 57.32±1.49 | 51.78±1.41 | 51.75±1.41 | 44.98±1.40 | 44.95±1.38 | **39.38±1.34** | **39.39±1.33** |
> | 10 | **57.67±1.63** | 57.92±1.58 | 52.01±1.55 | 51.99±1.54 | 45.35±1.53 | **45.25±1.52** | 39.19±1.44 | 39.31±1.40 |
> | 12 | 57.65±1.68 | **58.01±1.65** | **52.12±1.60** | **52.13±1.60** | **45.36±1.58** | 45.22±1.53 | 39.03±1.50 | 39.24±1.42 |
> | 14 | 57.49±1.73 | 57.96±1.66 | 52.08±1.66 | 52.06±1.65 | 45.26±1.63 | 45.20±1.55 | 38.81±1.55 | 39.10±1.48 |
> | 18 | 57.13±1.82 | 57.98±1.72 | 52.01±1.76 | 52.03±1.75 | 44.89±1.74 | 45.04±1.53 | 38.25±1.65 | 38.97±1.49 |
> | 22 | 56.65±1.92 | 57.89±1.72 | 51.81±1.87 | 51.80±1.85 | 44.46±1.85 | 45.01±1.53 | 37.77±1.77 | 38.83±1.51 |
> | 26 | 56.34±2.01 | 57.90±1.75 | 51.38±1.99 | 51.49±1.64 | 43.98±1.97 | 44.92±1.53 | 37.34±1.90 | 38.72±1.43 |
> | 30 | 56.10±2.13 | 57.92±1.79 | 50.82±2.13 | 51.38±1.75 | 43.59±2.11 | 44.90±1.65 | 37.06±2.05 | 38.66±1.45 |
> | 32 | 56.05±2.19 | 57.94±1.81 | 50.48±2.21 | 51.29±1.77 | 43.35±2.19 | 44.86±1.62 | 36.91±2.13 | 38.66±1.51 |
>
> **The observed reward hacking curve exactly matches our theoretical prediction in Theorem 1.** For example, Best-of-$n$ clearly demonstrates hacking on the GPQA dataset, even with the very capable Skywork-Llama 3.1 8B as proxy reward (ranked # 10 out of all non-generative reward models on RewardBench [2]). While performance initially climbs from $40.70\%$ at $n = 1$ to a peak of $45.36\%$ at $n = 12$, it ultimately decreases to $43.35\%$ as over-optimization sets in.
>
> **Hedging mitigates reward hacking in all cases.** Soft BoN delivers more stable and often superior performance than BoN. On GPQA, it reduces the performance drop from over 2 points to less than 0.4 points. On MATH, where hacking is most severe, it effectively mitigates BoN’s 2.5-point collapse. This stability also enables Soft BoN to achieve a higher peak score on MMLU (Llama-3 8B).
>
> The hedging benefit holds for learning from human preferences (RLHF), complex reasoning (MMLU), math (MATH), and science questions (GPQA), and is demonstrated with multiple large reward models (8B parameters). Please let us know if you have any additional questions about our setup or our results!
>
> **References:**
> 1.  Frick et al. *How to Evaluate Reward Models for RLHF*. ICLR 2025.
> 2.  Lambert et al. *RewardBench: Evaluating Reward Models for Language Modeling*. NAACL 2025.
>
> ---
>
> # W1: ***Need for true reward***
>
> You are correct that the need for true reward scores can be a practical barrier. We note that without a “true reward” to serve as a comparison (ground-truth) baseline, it is challenging to measure hacking. By definition, hacking occurs when a model over-optimizes to a proxy reward, and, consequently, precisely evaluating this phenomenon requires a reference ground-truth. Our experimental results on MMLU, MATH, and GPQA confirm that our framework offers a practical and effective tool to mitigate hacking in real-world reasoning tasks. However, we recognize this limitation in Line 234, and plan to re-iterate this in a revised conclusion section.
>
> ---
>
> ### Minor Labeling Issue:
>
> Thanks for catching this. We will fix the axis’ label before the final version.
>
> ---
>
> # Summary:
> We are glad that your detailed feedback has strengthened the paper on three key fronts. First, we have provided a new theoretical analysis of HedgeTune's sample complexity, formally proving its practical efficiency. Second, we have demonstrated the generality of our approach with extensive new experiments on complex reasoning benchmarks (MMLU, MATH, GPQA) using powerful 8B reward models. Finally, we have clarified the core limitation of our method while highlighting its value in high-stakes, verifiable domains. Thank you for your constructive guidance and let us know if you have any questions

---

### Official Review · Reviewer_du2S · 2025-07-03

**Clarity:** 3
**Significance:** 2
**Originality:** 3
**Rating:** 5
**Confidence:** 3

**Summary:**

Instead of using best-of-n sampling, a new strategy called best-of-Poisson is proposed and analyzed. Additionally, soft-best-of-n where instead of taking the mode of the distribution, the chosen outcome is sampled, is also considered. This helps avoiding overestimating (or underestimating) the proxy reward and tries to mitigate the problem of overoptimizing for the proxy reward in alignment.

**Questions:**

- A 'soft best-of-Poisson' seems like a natural extension of this work. Was it tried by any chance? Not having it does not affect the rating of the paper in any way. I am just curious.
- I wonder if the specified for this paper is correct. I will let other reviewers have their say. Although yes this is alignment and it relates to safety. But I see it more as general machine learning maybe? Because it's not directly related to safety. I am not sure. But this might be something to think about.

**Ethical Concerns:**

["NO or VERY MINOR ethics concerns only"]

**Final Justification:**

I have read all the discussions regarding this. I am divided between a 4 and a 5. (Borderline Accept vs Accept). My concerns are mostly addressed, and although the need for a validation set is pointed out by other reviewers (and the paper also explicitly mentions it), maybe this is useful when a validation set can be obtained. I am leaning towards accept conditional on clarifications and edits promised in the rebuttals. I am upgrading my rating to "5". My concerns about why it could not be a 5, but a 4 are only in terms of impact as the rating mentions "high impact on one sub-area of AI". My confidence score reflects this uncertainty. Other than I am happy with the technical evaluation, theoretical justifications, motivations, etc.

**Limitations:**

Limitations are not adequately explicitly discussed in my opinion.

**Paper Formatting Concerns:**

- Majority of the references (maybe ~80%) are arXiV preprints. I think many of these are also published in peer-reviewed venues. Maybe consider citing those instead?

**Quality:**

3

**Strengths And Weaknesses:**

**Strengths**
- Interesting new way of sampling that seems to avoid proxy reward hacking
- Decent experimental results
- Situated well with related work.

**Weaknesses**
- Introduction, paper begins by saying DPO aims to maximize reward function. DPO does not model a reward function and instead directly optimizes policy. Am I misunderstanding something?
- The 10^-3 gap is an empirical result under specific settings, but it is not clear from the text that it is. It reads like a theoretical result: "bounded by 10^-3".
- This is an online algorithm. Both online and offline algorithms are important in their own ways but this is both a strength and a weakness.
- Access to true rewards needed. But this limitation explicitly stated.

---

> ### Author Rebuttal · Authors · 2025-07-30
>
> Thank you so much for your helpful feedback and thoughtful reading of our work. We address below the questions and points raised in your review and would be excited to answer any additional questions.
>
> ---
> # W1: *Does DPO optimize a reward function?*
> Thanks for raising this point. BoN, RLHF, and DPO make use of reward functions/signals. These reward functions can be implicit or explicit. For instance, BoN uses an *explicit* reward function (e.g., a model that assigns scores to responses). In contrast, DPO optimizes policy under an *implicit* reward function by recognizing that the optimal policy in RLHF (defined in Eq. 1 in our work) is an exponential tilt of the reference policy. DPO simply inverts this relationship and defines the loss directly on the policy (hence a language model being "secretly" a reward model). We will clarify this in the paper.
>
> ---
>
> # W2: *The $10^{-3}$ KL gap as an empirical result*
> The $10^{-3}$ gap presented in Appendix B is indeed based on a numerical estimate under our assumptions. We first derive a closed-form expression of the KL gap (see Eq. 84) in terms of the Poisson parameter $\mu$. Next, we numerically find an upper bound of the expression (see Figure 2 for a plot) to demonstrate how negligible the gap is. For completeness, we will add a proof to the paper that shows *explicitly* why the bound holds. We added the proof to the end of our response so we would not disrupt the flow of our conversation. Please let us know if you have any questions about it.
>
>  ---
>
> # W3/W4: *Algorithmic nature and access to true rewards*
> You are correct that the need for true reward scores can be a practical barrier. Without a "true reward" to serve as a comparison (ground-truth) baseline, it is challenging to measure hacking. By definition, hacking occurs when a model over-optimizes to a proxy reward, and, consequently, precisely evaluating this phenomenon requires a reference ground-truth. We recognize this in Line 234 and plan to re-iterate it in a revised conclusion section, along with the point you raised on the online/offline nature of the method.
>
> ---
>
> # Q1: *‘Soft best-of-Poisson’ seems like a natural extension*
> We love this idea! After reading your review, we looked into this method and found that its properties can potentially extend our existing results, albeit at the cost of a more cumbersome mathematical analysis. We will include this as a blueprint for future research direction in the last section. We will add to the appendix: (i) the SBoP algorithm, (ii) its expected reward, and (iii) the KL divergence with respect to the reference policy. As one might expect, these correspond exactly to the expressions for Soft Best-of-$n$, but averaged over a Poisson distribution.
>
> We find your question particularly interesting since it highlights our intention behind studying Best-of-Poisson and Soft Best-of-$n$. We wanted to leverage two distinct control mechanisms that inference-time alignment offers us: control over the generation through $n$ (be it deterministic or randomized) and control over the selection through the temperature $\lambda$. Extensions such as SBoP show that these methods can be combined, offering richer control over the alignment process. Thank you for prompting this interesting discussion!
>
>  ----
>
> # Q2: *Paper specified area*
> We do recognize that our work is largely technical. AI safety concerns are, at their heart, *socio-technical* [1]. The impact of model failures due to hacking (and beyond) will depend on the application and stakeholders at hand. One clear example is an AI agent generating controversial and toxic content on X (previously Twitter) to optimize its engagement metric at the direct expense of content quality and safety [2]. Other dangerous patterns of failure include models that learn to be sycophantic instead of truthful [3] or actively exploit loopholes to circumvent oversight [4]. The goal of our work was to study the phenomenon of hacking from a rigorous mathematical perspective -- which can help with the design of hacking mitigation (and hence AI safety methods) with provable performance guarantees. We find technical approaches such as ours important for AI safety in practice, although recognize that they must exist within a larger safety ecosystem where rigorous technical foundations inform responsible deployment practices and policy decisions, and vice-versa. We will add this discussion to the last section of the paper.
>
> ### References:
> 1.  Amodei et al. "Concrete Problems in AI Safety." 2016.
> 2.  Pan et al. "Feedback loops with language models drive in-context reward hacking." ICML 2024.
> 3.  Sharma et al. "Towards Understanding Sycophancy in Language Models." ICLR 2024.
> 4.  OpenAI. "Detecting misbehavior in frontier reasoning models." Blog post 2025.
>
> ---
>
> # *Limitations*
>  Thanks -- we will improve this. We will rename our conclusion section to "Future Work and Limitations" and further highlight the points raised by the reviewer:
> 1.  Need for true reward model to measure and control hacking as *stated in Line 235*;
> 2.  Algorithmic advantages and limitations in terms of the online/offline nature;
> 3.  Richer hedging policies (e.g., "Soft Best-of-Poisson");
> 4.  How this work can be used in AI safety practices and deployment
> Thanks again for this suggestion.
>
> ---
>
> ### Majority of the references are arXiV preprints:
>
> This was an oversight on our part during the preparation of the manuscript, and we will fix it promptly!
>
> ---
>
> # Summary:
>
> In summary, your feedback has prompted us to strengthen the paper on several key fronts. We have replaced a numerical claim with a formal proof, suggested new inference-time methods for future work, and elaborated on the limitations of our work and its impact to the AI safety community. We believe these additions make a much stronger case for our contributions, and we thank you for your guidance and time!
>
> ---
>
> # Proof of the KL gap:
>
> Let the following two functions
> $$
> q_{\mu}(x) = \bigl(1+\mu x\bigr) e^{\mu(x-1)},
> \qquad
> g_{\lambda}(x) =\frac{\lambda e^{\lambda x}}{e^{\lambda}-1},
> \qquad 0\le x\le1,
> $$
> denote, respectively, the BoP density with the Poisson rate $\mu$ and the exponential‑tilt density with parameter $\lambda$. For any $\mu > 0$, the value $\lambda^\ast=\lambda^\ast(\mu)$ is chosen so that the two laws have the same first moment (i.e., same expected reward). We first note the expected reward of the BoP policy lies in the interval $(\frac{1}{2}, 1)$. Additionally, the tilted expected reward is a strictly increasing function of $\lambda$ over the same range since its derivative is the (positive) variance of the reward under the tilted distribution. By the Intermediate Value Theorem, there exists a unique $\lambda^* = \mu + \delta(\mu)$, where $\delta(\mu) = \lambda^*(\mu) - \mu$, such that the two rewards are equal.
>
> Because $g_{\lambda(\mu)}$ is the information projection of $q_{\mu}$ onto the exponential family $\{g_{\lambda}\}_{\lambda>0}$ under the linear reward-matching constraint, the Csiszár–Pythagoras identity gives
>
> $$D_{\text{KL}}(q_\mu || \pi_{\text{ref}}) - D_{\text{KL}}(g_{\lambda(\mu)} || \pi_{\text{ref}}) = D_{\text{KL}}(q_\mu || g_{\lambda(\mu)})$$
>
> To bound this KL divergence, we use a general inequality which we derive below from the properties of the likelihood ratio function. Let $L(x) = \frac{p(x)}{q(x)}$ denote the likelihood ratio between two distributions $p$ and $q$. We first define the KL divergence in terms of the convex function $\varphi(t) = t \log t - t + 1$ and then note that the expectation of any function is upper bounded by its essential supremum. We can retrieve the following upper bound:
>
> $$D_{\mathrm{KL}}(p || q) =  E_q[\varphi(L(x))]  \le \sup_{x} \varphi(L(x)) = \sup_{t \in \mathrm{Im}(L)} \varphi(t)  $$
>
> In our problem, the likelihood ratio is given by
>
> $$L_\mu(x) =  \frac{q_\mu(x)}{g_{\lambda^*}(x)} =  \frac{(\mu x + 1)(e^{\lambda^\ast} - 1)}{\lambda^\ast e^{\mu}}  \cdot e^{-\delta \mu x}$$
>
> To obtain a uniform bound on $L_\mu(x)$, define
>
> $$\alpha = \sup_{\mu > 0, x \in [0,1]} \left| \log L_\mu(x) \right|$$
> This is a well-posed, nested optimization problem requiring numerical solution. The procedure is as follows:
> 1.  For a given $\mu$, numerically solve for the unique $\lambda^*$ satisfying the reward-matching equation.
> 2.  For the resulting pair $(\mu, \lambda^*)$, compute $\max_{x \in [0,1]} |\log L_\mu(x)|$ by evaluating the boundary values and any interior critical points.
> 3.  Maximize the result of step (2) over all $\mu > 0$.
>
> This procedure is deterministic, and its solution yields a uniform bound:
> $$
> e^{-\alpha} \leq L_\mu(x) \leq e^{\alpha}, \quad \text{for all } \mu > 0 \text{ and } x \in [0,1] \quad \text{with }\alpha \approx 0.03968
> $$
> Since the function $\varphi(t)$ is convex, we verify that its supremum over $[e^{-\alpha}, e^{\alpha}]$ is attained at an endpoint:
> $$
> D_{\text{KL}}(q_\mu || g_{\lambda(\mu)}) \leq \varphi(e^\alpha) = \alpha e^\alpha - e^\alpha + 1 \approx 0.03968 \cdot e^{0.03968} - e^{0.03968} + 1 \approx 8 \times 10^{-4}
> $$
> This bound is independent of $\mu$, and therefore holds uniformly.

---

> > ### Comment · Reviewer_du2S · 2025-08-09
> >
> > I am very thankful to the authors for the detailed response. Regarding w1, in my experience I have seen the word 'reward' alone only being used when the reward is explicitly computed, but I am thankful for the clarification.
> >
> > My concerns were addressed, and I have no more comments.
> >
> > I have read the discussions with other reviewers as well, and think the authors have adequately addressed the concerns.
> >
> > Although the need for true reward is a weakness, maybe this method is useful when a certain sample from the population can be used to prevent reward hacking.

---

### Note · Authors · 2025-08-13

We thank the reviewers again for the detailed and constructive conversation, as well as the Area Chair for their time and effort.

We are pleased that the reviewers acknowledged we addressed all their questions in detail. As a result of the discussion period, we have sharpened our paper on many fronts. First, we demonstrated the generality of our hedging framework in setups with verifiable rewards, such as math or reasoning benchmarks. Second, we extended our experimental setup to include larger and more powerful reward models. Moreover, we clarified the intuition behind our technical assumptions, thanks to a question from Reviewer 51Rg, and incorporated additional edits to enhance the paper’s presentation.

We are grateful for the opportunity to improve our work based on this feedback and will incorporate all discussed revisions into the camera-ready version.

With best regards,

— The Authors of Paper 25917

---

### Decision · Program_Chairs · 2025-09-17

**Decision:**

Accept (spotlight)

**Comment:**

This paper provides a formal characterization of reward hacking in inference-time alignment for large language models. The authors prove that under mild technical assumptions, optimizing for a proxy reward inevitably leads to a degradation in true performance after a certain threshold. To mitigate this, they introduce "hedging" as a strategy, propose a novel sampling method called Best-of-Poisson (BoP), and develop an efficient algorithm, HedgeTune, to find the optimal inference-time parameter that maximizes true reward without over-optimizing. Its core scientific contribution is the formal characterization, under mild assumptions, of how optimizing for a proxy reward inevitably leads to a decline in true performance, following a predictable unimodal curve.

The strengths, as noted by reviewers (pDYx, Eaod), lie in this strong theoretical foundation and the proposal of novel, practical mitigation methods like Best-of-Poisson and HedgeTune (du2S, 51Rg). The principal weakness, unanimously identified, is the method's reliance on a validation set with access to true rewards, which may limit its applicability (pDYx, Eaod). However, the authors' rebuttal provided substantial new experiments on complex reasoning benchmarks with large models, directly addressing initial concerns about the work's empirical scope and demonstrating its relevance. This, combined with a new analysis of sample complexity, makes a compelling case for the paper's value in settings where a limited set of ground-truth evaluations is feasible. The work stands out for its theoretical elegance and demonstrated practical utility, justifying its acceptance.